# DoReMi: Grounding Language Model by Detecting and Recovering from Plan-Execution Misalignment

## Abstract

Large language models (LLMs) encode a vast amount of semantic knowledge and possess remarkable understanding and reasoning capabilities. Previous work has explored how to ground LLMs in robotic tasks to generate feasible and executable textual plans. However, low-level execution in the physical world may deviate from the high-level textual plan due to environmental perturbations or imperfect controller design. In this paper, we propose **DoReMi**, a novel language model grounding framework that enables immediate **D**etection and **Re**covery from **Mi**salignments between plan and execution. Specifically, we leverage LLMs to play a dual role, aiding not only in high-level planning but also generating constraints that can indicate misalignment during execution. Then vision language models (VLMs) are utilized to detect constraint violations continuously. Our pipeline can monitor the low-level execution and enable timely recovery if certain plan-execution misalignment occurs. Experiments on various complex tasks including robot arms and humanoid robots demonstrate that our method can lead to higher task success rates and shorter task completion times. Videos of DoReMi are available at `https://sites.google.com/view/doremi-paper`.

## 1 Introduction

Large language models (LLMs) pre-trained on web-scale data emerge with common-sense reasoning ability and understanding of the physical world. Previous works have incorporated language models into robotic tasks to help embodied agents better understand and interact with the world to complete challenging long-horizon tasks that require complex planning and reasoning (Ahn et al., 2022; Huang et al., 2022a; Liang et al., 2022).

To make the generated plan executable by embodied agents, we need to ground the language. One line of the works leverages pre-trained language models in an end-to-end manner that directly maps language and image inputs to the robot's low-level action space (Brohan et al., 2022; 2023; Jang et al., 2022; Shridhar et al., 2023; Nair et al., 2022). These approaches often require large amounts of robot action data for successful end-to-end training, which is expensive to acquire (Brohan et al., 2022). Moreover, these action-output models often contain large transformer-based architectures and cannot run at high frequencies. Therefore, they may not be suitable for tasks with complex dynamics (e.g., legged robots) that require high-frequency rapid response. Recently, many works have adopted a hierarchical approach where language models perform high-level task planning, and then some low-level controllers are adopted to generate the complex robot control commands (Ahn et al., 2022; Huang et al., 2022a; Liang et al., 2022; Huang et al., 2022b). Under this hierarchical framework, we can leverage powerful robot control methods, such as reinforcement learning, to handle complex robot dynamic control problems with high frequency.

However, these grounding methods often assume that every low-level skill can perfectly execute the high-level plan generated by the language model. In practice, low-level execution may deviate from the high-level plan due to environmental perturbations or imperfect controller design. These misalignments between plan and execution may occur at any time during the task procedure. Previous works consider incorporating execution feedback into language prompts once the previous plan step is finished. If the step is unsuccessful, the process is repeated (Huang et al., 2022b). However, this

delayed feedback can be inefficient. For instance, as illustrated in Figure 1(b), when a human is carrying a box and performing the low-level skill "Go to the gray table", if the box is accidentally dropped, it becomes futile to continue with the current skill. The human will immediately abort the current skill and call for the skill "Pick up the box". However, agents without immediate re-planning will continue going forward and will take more time to pick up the box dropped halfway after reaching the destination.

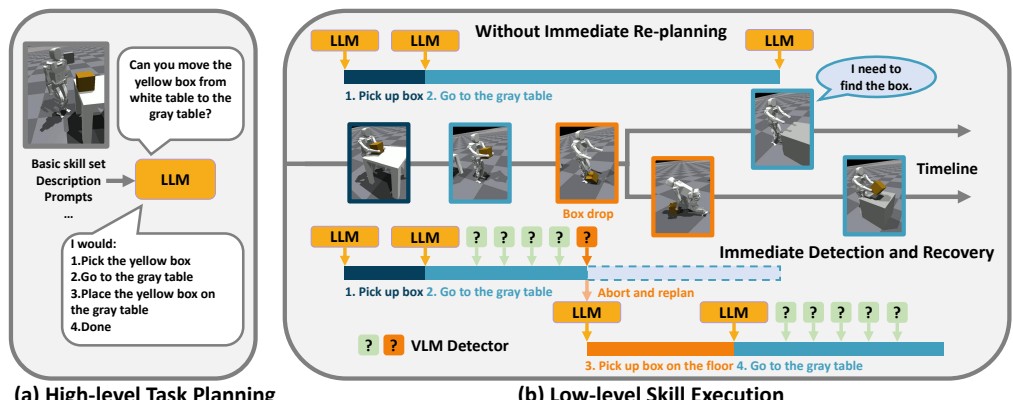

Figure 1: Illustration of our motivation. Low-level execution may deviate from the high-level plan. DoReMi can immediately detect the misalignment between the plan and execution when the box drops accidentally and quickly recovers. Agents without immediate re-planning suffer from such misalignment.

In this paper, we propose a novel framework **DoReMi** which enables immediate **De**tection and **Re**covery from plan-execution **Mi**salignments. Specifically, in addition to employing LLMs for high-level planning (Ahn et al., 2022), we further leverage LLMs to generate constraints for low-level execution based on their understanding of physical worlds. During the execution of low-level skills, a vision language model (VLM) (Li et al., 2023b) is employed as a general "constraint detector" to monitor whether the agent violates any constraints continuously. If some constraints are violated, indicating that the plan and execution may be misaligned, the language model is immediately called to re-plan for timely recovery. We summarize several advantages of our pipeline: (1) LLM plays a dual role, aiding not only in high-level planning but also in supervising low-level execution, enabling rapid detection and recovery; (2) The VLM can focus on the specific constraints suggested by the LLM and only need to pick binary answers, providing more precise feedback. This collaborative approach between the LLM and the VLM can help align the plan and execution during the whole task period. Furthermore, under mild assumptions, we conduct a theoretical analysis to estimate how much time can be saved or how much the success rate can be improved through immediate re-planning when misalignment occurs. Experiments in physical simulations, including robot arm manipulation tasks and humanoid robot tasks, demonstrate that DoReMi leads to a higher task success rate and shorter task execution time.

## 2 RELATED WORKS

**Language Grounding** Prior research has attempted to employ language as task abstractions and acquired control policies that are conditioned on language (MacMahon et al., 2006; Chaplot et al., 2018; Jiang et al., 2019a; Misra et al., 2017; Mei et al., 2016). Furthermore, some studies have investigated the integration of language and vision inputs within embodied tasks to directly predict the control commands (Silva et al., 2021; Guhur et al., 2023; Goyal et al., 2021). Recent works, including (Brohan et al., 2022; 2023; Shridhar et al., 2023; Zhang & Chai, 2021; Lynch et al., 2022), have demonstrated significant progress in utilizing transformer-based policies to predict actions. However, these end-to-end approaches heavily depend on the scale of expert demonstrations for model training.

**Task Planning with Language Model** Traditionally, task planning was solved through symbolic reasoning (Nau et al., 1999; Fikes & Nilsson, 1971) or rule-based planners (Fox & Long, 2003; Jiang et al., 2019b). Recently, many works demonstrated that large language models (LLMs) can generate

executable plans in a zero/few-shot manner with appropriate grounding (Huang et al., 2022a; Ahn et al., 2022; Zeng et al., 2022; Ren et al., 2023). Some pre-trained low-level skills (primitives) are then adopted to execute steps in order. These LLM planners typically assume the successful execution of each skill, resulting in an open-loop system in physical worlds. Works in the instruction-following benchmark (Shridhar et al., 2020; Puig et al., 2018) like ReAct (Yao et al., 2022), and Reflexion (Shinn et al., 2023), incorporate feedback into LLM prompts to help planning after each step of the plan is finished. However, these benchmarks operate in discrete scenes and pay less attention to the skill execution period. The closest work to ours is Inner Monologue (Huang et al., 2022b), which also considers continuous physical worlds, and takes into account 3 types of feedback (e.g. success detectors, scene descriptions, and human feedback) upon the completion of each step. However, Inner Monologue's feedback is impractical and hard to obtain at high frequency. In contrast to this, our framework enables precise and high-frequency feedback with practical detectors.

**Vision Language Model for Embodied Control.** The vision language model (VLM) is trained on image-text pairs, enabling it to simultaneously understand visual and textual inputs and address a variety of downstream tasks, such as visual question answering (VQA)(Li et al., 2023b; Antol et al., 2015), image captioning (Zhou et al., 2020), and object detection (Gu et al., 2021). VLMs align semantic information between vision and natural language, thereby aiding in grounding language models and facilitating embodied control. Pre-trained visual encoders or instruction encoders (Radford et al., 2021) can be connected with some action head to help train end-to-end policies (Shridhar et al., 2022) or generate textual plans (Driess et al., 2023). RT-2 (Brohan et al., 2023) directly fine-tuned on a VLM can generate texts and robot control actions simultaneously. VLMs can also act as scene descriptors(Huang et al., 2022b), success detectors (Du et al., 2023; Zhang et al., 2023), or object detectors(Stone et al., 2023) to facilitate the task execution. To ensure adherence to crucial constraints, we employ the VLM (Li et al., 2022) as a "constraint detector", periodically verifying whether the agent satisfies specific constraints.

## 3 PROBLEM STATEMENT

Our objective is to enable the embodied agent to accomplish long-horizon tasks specified as natural language instructions $i$ in the physical world. The agent has a basic skill set $\Pi$, with each skill $\pi_j \in \Pi$ corresponding to a distinct function that can be described in natural language $l_{\pi_j}$.

Previous work has illustrated that pre-trained large language models can be used as planners to decompose complicated language instructions into textual skill sequences: $i \rightarrow (l_{\pi_1}, l_{\pi_2}, ..., l_{\pi_n})$ (Huang et al., 2022a; Zeng et al., 2022), as shown in Figure 2a. Many works consider feedback at the end of each skill (Huang et al., 2022b; Yao et al., 2022; Shinn et al., 2023), which can be described as *plan-level feedback* in Figure 2b. In particular, Inner monologue (Huang et al., 2022b) assumes the accessibility of 3 sources of oracle feedback from success detectors, passive scene descriptors, and humans. However, such oracle feedback is impractical in most settings and can not be frequently obtained: the success detector can only assess success or failure upon the completion of each skill, humans are unable to provide high-frequency feedback, and frequently injecting passive scene descriptions into the LLMs risks exceeding its maximum input token length and may cause a performance drop in LLMs (Liu et al., 2023). *How to incorporate frequent and precise feedback into the LLMs remains a challenge.*

In the following section, we will introduce our DoReMi framework which leverages powerful LLMs to generate both high-level plans and low-level execution constraints, which then enables *execution-level feedback* by VLM during the entire execution period, as shown in Figure 2(c).

## 4 METHOD

In this section, we introduce our **DoReMi** framework which enables immediate **De**tection and **Re**covery from Plan-Execution **Mi**salignment. Our algorithm can be succinctly described in two stages depicted in Figure 2(c):

1. At the high-level planning stage, given a set of low-level skills, prompts, and high-level task instruction, language models are leveraged to play a dual role, aiding not only in planning the next skill but also generating constraints for the next skill based on historical information.

2. During the low-level skill execution stage, we employ a vision-language model (VLM) (Li et al., 2023b) as a general "constraint detector" that periodically verifies the satisfaction of

all constraints. If any constraint is violated, the language model is invoked for immediate re-planning to facilitate recovery.

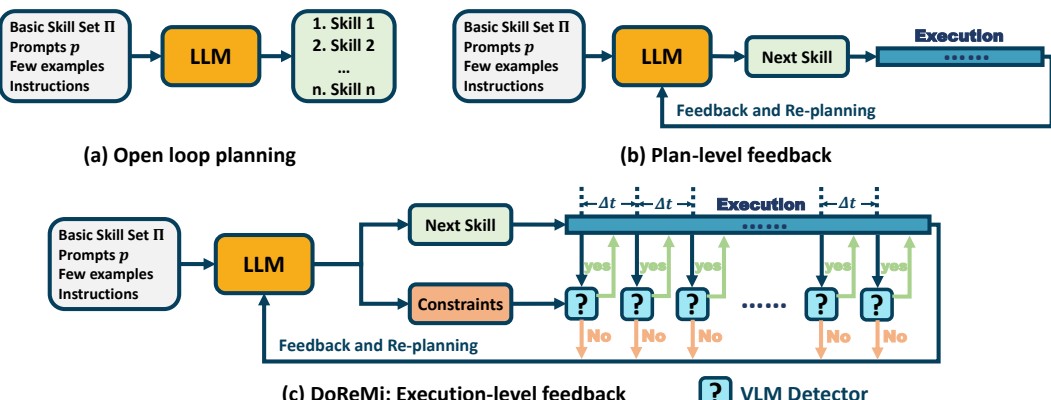

Figure 2: Previous methods perform open-loop planning or only re-plan when the previous skill is finished. Our DoReMi framework leverages LLM to generate both the plan and corresponding constraints. Then a VLM is employed to supervise the low-level execution period, which enables immediate recovery from plan-execution misalignment.

### 4.1 LANGUAGE MODEL FOR PLANNING

Following previous works that leverage LLM to generate feasible textual plans(Ahn et al., 2022), we utilize LLMs to plan the next steps through few-shot in-context learning. Furthermore, we employ language models for re-planning when our constraint detector identifies a plan-execution misalignment. In such scenarios, we additionally include the misalignment information in prompts and invoke the LLM for re-planning. Detailed planning prompts can be found in Appendix D. Practically, we deploy the Vicuna-13B model (Chiang et al., 2023) locally and pick the next skill with max output probability. We also try GPT4 (OpenAI, 2023) through OpenAI API to directly output the next step with zero temperature. Both LLMs exhibit effective planning capabilities in our tasks.

### 4.2 LANGUAGE MODEL FOR CONSTRAINT GENERATION

LLM planner helps agents decompose long-horizon tasks into skill sequences. However, LLMs are not inherently integrated into the execution of low-level skills, which potentially leads to misalignment between plan and execution. To further explore the ability of LLMs in embodied tasks, we utilize LLMs not only for next-step planning but also for constraint generation based on historical information. For instance, consider the execution period of the *"go to"* skill after the *"pick up box"* skill. In such cases, the constraint *"robot holds box"* must be satisfied and violation of this constraint could indicate a failure in the picking or possible dropping of the box. Similarly, after the skill *"place red block on green block"*, the constraint *"red block on green block"* should always be met. LLMs possess the capability to automatically generate these constraints for planned steps, drawing upon their encoded understanding of the physical world. Moreover, the VLM detector can focus on these specific constraints and only need to pick binary answers from "Yes" or "No", resulting in much more precise feedback. In contrast, open-ended scene descriptions of VLMs may result in large ambiguity and miss essential information, as shown in Figure 3.

In practice, after the LLM selects the next step with the highest output probability, we continue the generation starting with "Constraint:" to derive specific constraints. Additionally, we conducted experiments to assess the quality of the LLM-generated constraints. First, we conduct a user study to compare the LLM-generated constraints with manually specified constraints. Survey results show that users think 98% of the LLM-generated constraints are reasonable and admissible. Second, We query VLM with manually specified constraints and LLM-generated constraints respectively, picking binary answers from {"Yes", "No"}. We find these two answers are the same in 97% of the queries. These results show the remarkable proficiency of LLMs in generating constraints, driven by their encoded understanding of the physical world. For a more comprehensive analysis, please refer to Appendix D.

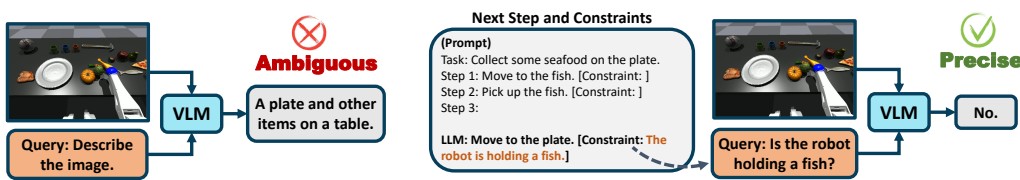

(a) Open-ended scene description  (b) VLMs focus on specific constraint generated by LLMs

Figure 3: Open-ended scene descriptions of VLMs are ambiguous. DoReMi leverages the LLM to generate specific constraints for steps and directly queries the VLM with these constraints, resulting in much more precise feedback.

---

**Algorithm 1 DoReMi** (Immediate **De**tection and **Re**covery from **Mi**salignment)

**Given:** A high level instruction $i$, a skill set $\Pi$, language description $l_\Pi$ for $\Pi$, language model $L$, prompt $p_0$, and VLM constraint detector $D$.

1: Initialize the skill sequence $\pi \leftarrow \emptyset$, the number of steps $n \leftarrow 1$.
2: **while** $l_{\pi_{n-1}} \neq done$ **do**
3:    $\pi_n \leftarrow \arg\max_{\pi \in \Pi} L(l_\pi | i, p_{n-1}, l_{\pi_{n-1}}, ..l_{\pi_0}), c_n \leftarrow L(i, p_{n-1}, l_{\pi_n}, ..l_{\pi_0})$
4:    Update prompt $p_n$.
5:    **while** $\pi_n$ is not finished **do**
6:       Every $\Delta t$ second, query agent all the constraints $c_n$ using the constraint detector $D$.
7:       **if** $\exists D(c_n) = $ **false then**
8:          Add constraint violate information into prompt $p_n$ and **break**.
9:       **end if**
10:    **end while**
11:    $n \leftarrow n + 1$.
12: **end while**

---

### 4.3 VLM AS CONSTRAINT DETECTOR

Subsequent to the constraint generation stage, the agent proceeds to execute the planned step while adhering to constraints suggested by the LLM. The LLM-generated constraints may include various types, such as "red block is on blue block," "no obstacles in front of the robot," "robot is holding an apple," and more. In this work, we adopt a vision language model(VLM) (Li et al., 2023b) as a general "constraint detector" to check all constraints through visual information. The visual input of the VLM is captured from either a first-person or third-person perspective camera, and the text input is automatically adapted from the LLM proposed constraints in the form "Question: Is the *constraint $c_j$ satisfied*? Answer:". For each query, the VLM only needs to select an answer from {"Yes", "No"}, which consists of very short token lengths and costs less than 0.1 second. We use $D(c_j)$ to denote the answer of the VLM $D$ when checking constraint $c_j$. If $c_j$ is satisfied, $D(c_j) = True$; otherwise, $D(c_j) = False$. The pseudo-code of the pipeline is provided in Algorithm 1. It's also worth mentioning that detectors in other modalities are also compatible with our framework and constraint detectors can run parallel to low-level controllers with different frequencies.

In practice, we use the pre-trained BLIP-2 model (Li et al., 2023b) as a general "constraint detector" to periodically check whether the agent satisfies all constraints every $\Delta t = 0.2$ second. If so, the robot continues executing the current low-level skill; otherwise, the robot aborts the current skill, and the re-planning process is triggered. We observe that pre-trained zero-shot VLM can perform well in most tasks, except those with extremely complex scenes. To enhance the performance in such complex tasks, we collect a small dataset and fine-tune the VLM using the parameter-efficient LoRA method (Hu et al., 2021). We also verify that the fine-tuned VLM detector can generalize to unseen objects, unseen backgrounds, and even unseen tasks.

### 4.4 THEORETICAL ANALYSES

Delayed re-planning may waste time (as shown in Figure 1) or even result in failures. In this section, we analyze the potential time savings and success rate improvements achievable through immediate detection and recovery. We denote the execution time of low-level skill with random variable $t$ with mean $\mathbb{E}[t] = \mu$ and variance $Var(t) = \sigma^2$. Misalignment can occur at any time $s$ within the

execution time interval $[0, t]$ where $0 \leq s \leq t$. Additionally, we assume our constraint detector has probability $p_d$ to detect each misalignment. We define the discrete random variable $M$ as the number of misalignment occurrences under the following assumptions: (1) Plan-execution misalignments occur independently. (2) Misalignments occur at a constant ratio $\lambda$ within a small time interval: $\lim_{t \to 0} P(M = 1) = \lambda t$. (3) No two misalignments occur simultaneously: $\lim_{t \to 0} P(M = k) = 0$ for $k > 1$. Under these assumptions, the number of plan-execution misalignments follows a Poisson distribution (Papoulis & Unnikrishna Pillai, 2002):

$$P(M = k) = \frac{(\lambda t)^k e^{-\lambda t}}{k!} \quad k = 0, 1, 2, 3... \tag{1}$$

**Theorem 1** *The following equations describe the possible time-savings $t_s$ and the success rate improvement $P_s$ under immediate detection and re-planning:*

$$\mathbb{E}(t_s) = \sum_k P(M = k) \mathbb{E}(t_w | M = k) = \frac{p_d \lambda (\mu^2 + \sigma^2)}{2} - p_d \lambda \mu \Delta t \tag{2}$$

$$\mathbb{E}(P_s) = 1 - \mathbb{E}(e^{-\lambda t}) \approx p_d \lambda \mu - \frac{(2p_d - p_d^2)\lambda^2(\mu^2 + \sigma^2)}{2} \tag{3}$$

The detector's reaction time, $\Delta t$, is much smaller than the average execution time $\mu$, so time-saving $\mathbb{E}(t_s)$ is greater than 0. $\lambda$ represents the misalignment occurrence ratio per second, which is very small, so success rate improvement $\mathbb{E}(P_s)$ is also greater than 0. Detailed proof can be found in Appendix A.

## 5 EXPERIMENTS

In this section, we conduct experiments involving both robotic arm manipulation tasks and humanoid robot tasks, as shown in Figure 4. These tasks incorporate various environmental disturbances and imperfect controllers, such as random dropping by the robot end-effector, noise in end-effector placement positions, failure in pick, and unexpected obstacles appearing in the robot's path.

We aim to answer the following questions: (1) Does **DoReMi** enable immediate detection and recovery from plan-execution misalignment? (2) Does **DoReMi** lead to higher task success rates and shorter task execution time under environmental disturbances or imperfect controllers?

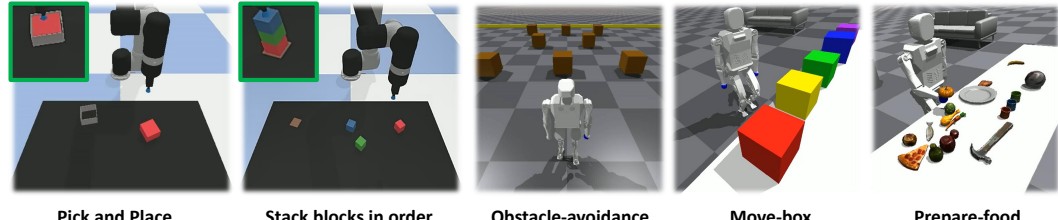

Figure 4: Robot manipulation and humanoid robot tasks in our experiments. We consider various types of environmental disturbance and imperfect controllers.

### 5.1 ROBOT ARM MANIPULATION TASKS

**Robot and Environment** This environment is adapted from *Ravens* (Zeng et al., 2020), a benchmark for vision-based robotic manipulation focused on pick-and-place tasks. An UR5e robot equipped with a suction gripper operates on a black tabletop, while a third-view camera provides a comprehensive view of the tabletop. The robot possesses a basic skill set including *"pick obj"* and *"place obj on receptacle"*, both of which are pre-trained primitives conditioned on single-step instructions similar to the CLIPort (Shridhar et al., 2022) and Transporter Nets (Zeng et al., 2020). To assess the effectiveness of our algorithm, we introduce additional disturbances into the original environment and the robot controller.

**Tasks: (1) Pick and Place.** The agent is required to pick a certain block and place it in a fixture. We assume the block has a probability $p$ to drop every second when sucked by the end-effector, so the agent may need to perform pick and place several times to finish the task. **(2) Stack blocks in order.**

| Tasks with disturbance | | Success Rate(%) ↑ | | | | | Execution Time(s) ↓ | | |
| --- | --- | --- | --- | --- | --- | --- | --- | --- | --- |
| | | SayCan | CLIPort | IM | DoReMi (ours) | IM-Oracle | IM | DoReMi (ours) | IM-Oracle |
| **Pick and place with random drop $p$** | $p$=0.0 | 100(±0) | 100(±0) | 100(±0) | 100(±0) | 100(±0) | **2.7**(±0.0) | **2.7**(±0.0) | 2.7(±0.0) |
| | $p$=0.2 | 81(±9) | 100(±0) | 100(±0) | 100(±0) | 100(±0) | 3.4(±0.2) | **3.0**(±0.2) | 3.4(±0.2) |
| | $p$=0.3 | 63(±9) | 100(±0) | 100(±0) | 100(±0) | 100(±0) | 4.0(±0.2) | **3.3**(±0.2) | 4.0(±0.2) |
| **Stack in order with noise $n$** | $n$=0.0 | **100**(±0) | **100**(±0) | **100**(±0) | **100**(±0) | 100(±0) | **7.2**(±0.0) | **7.2**(±0.0) | 7.2(±0.0) |
| | $n$=1.0 | 96(±4) | 96(±4) | 96(±4) | **100**(±0) | 100(±0) | 8.0(±3.0) | **7.5**(±0.5) | 7.4(±0.5) |
| | $n$=2.0 | 63(±9) | 85(±7) | 86(±7) | **96**(±4) | 98(±2) | 12.2(±5.3) | 10.2(±1.7) | 9.8(±2.0) |
| | $n$=3.0 | 31(±11) | 74(±10) | 75(±8) | **86**(±8) | 91(±7) | - | 15.6(±3.2) | 14.7(±2.3) |
| **Stack in order with noise $n$ random drop $p$=0.1** | $n$=0.0 | 71(±9) | 94(±7) | 94(±6) | **98**(±4) | 99(±1) | 10.0(±3.6) | **9.4**(±1.7) | 9.9(±1.9) |
| | $n$=1.0 | 71(±9) | **94**(±7) | 94(±7) | **94**(±7) | 97(±2) | 10.7(±3.9) | **10.6**(±3.2) | 10.9(±3.0) |
| | $n$=2.0 | 54(±12) | 79(±9) | 79(±8) | **92**(±6) | 95(±3) | - | **14.5**(±3.4) | 15.3(±3.5) |
| | $n$=3.0 | 21(±9) | 33(±10) | 34(±10) | **55**(±10) | 64(±8) | - | - | - |

Table 1: Success rates and task execution time under different degrees of disturbances. We only measure execution time under high success rates. The results show the mean and standard deviation over 4 different seeds, each with 12 episodes.

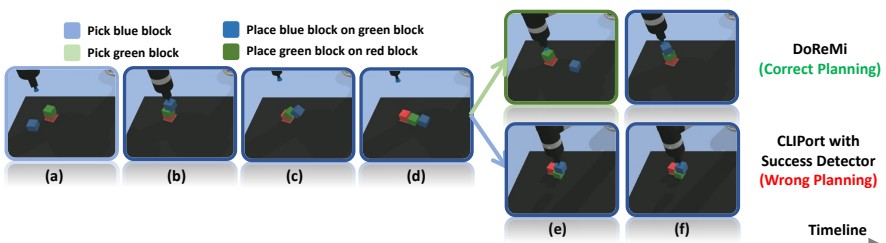

Figure 5: A comparison example. The robot arm tries to finish the step "Place blue block on green block" but collapses (bcd). DoReMi detects this misalignment and replans to pick and place the green block first (e). The baseline continues to repeat the previous step (ef) and results in failure.

The robot is required to stack several blocks in an order given by language instructions. The agent must perform "pick" and "place" skills in a precise sequence to successfully accomplish the task. We assume the controllers are not perfect by introducing uniform $[0, n]$ cm noise to the place positions. There is also a probability $p$ that a block held by the end-effector might randomly drop every second. The max execution time for all tasks is set to 20 seconds. Any execution that takes time longer than 20 seconds is considered as failure.

**Experiment Details** Following the pipeline in Figure 2, we use Vicuna-13B (Chiang et al., 2023) as LLM planner and zero-shot transferred BLIP-2 (Li et al., 2023b) as VLM constraint detector. We compare **DoReMi** with 4 baselines: **(1) SayCan**: an LLM is utilized to decompose instructions into steps and execute them sequentially. However, this approach assumes the successful execution of each step without considering potential failures. **(2) CLIPort**: a multi-task CLIPort policy conditioned on the single pick-place step. It utilizes an LLM to decompose instructions into steps and repeat each step until success. The same VLM is leveraged as a success detector to determine whether the current step should be repeated. **(3) Inner Monologue (IM)**: The same VLM is employed as scene descriptors and success detector to help LLM re-plan upon completion of each step. **(4) IM-Oracle:** Inner-Monologue with oracle feedback which does not exist in practical real-world settings. Results are shown in Table 1.

**Result Analyses** In the presence of disturbances, SayCan consistently fails in all tasks due to its lack of success detectors and re-planning mechanisms. In simple pick-place tasks, CLIPort and Inner-monologue with success detector can repeat the step and recover. However, they do not have a mechanism to abort the current execution and only re-plan at the end of each skill, resulting in a longer execution time. In the stack-block task, when encountering situations that require re-planning (e.g., the blocks collapse), CLIPort that only repeats the previous step fails to recover, as shown in Figure 5. When provided with imperfect scene descriptors (VLM), Inner Monologue also struggles to recover due to ambiguous open-ended scene descriptions. In contrast, DoReMi leverages LLMs to propose specific constraints for every low-level skill, with the VLM focused on these constraints, leading to highly accurate feedback. Furthermore, our VLM continuously detects constraint violations throughout the execution period, which enables immediate re-planning and recovery. Under these two mechanisms, DoReMi reaches higher success rates and shorter execution times.

## 5.2 HUMANOID ROBOT TASKS

**Robot Description and Low-level Skill Set** The humanoid robot utilized in our experiments possesses 6 degrees of freedom per leg and 4 degrees of freedom per arm, totaling 20 degrees of freedom. We equip the robot with a first-view camera on its base to provide visual information. Controlling complex humanoid robots with a single policy is challenging. Following the framework in Ma et al. (2022), we employ reinforcement learning to train the locomotion policy and leverage model-based controllers to acquire the manipulation policy. Specifically, we utilize the Deepmimic algorithm (Peng et al., 2018) to train a policy conditioned on commanded linear and angular velocity, allowing the robot to execute low-level skills such as "go forward 10 meters," "move forward at speed $v$," "go to *target place*," "turn right/left," and more. As for the manipulation policy, physically picking up objects is a challenging task, and we introduce an assistant pick-primitive similar to Li et al. (2023a), which can suck objects close to the end-effector. This enables the robot to execute low-level skills like "pick up *object*" and "place *object* on *receptacle*". Detailed architecture and training process can be found in Appendix B.

### 5.2.1 TASK CATEGORIES

We consider 3 categories of tasks and set the max task execution time to 90 seconds.

**(1) Obstacle-avoidance.** The robot performs the skill "go forward" to reach a finish line located at various distances. However, unknown obstacles may appear on the way with density $d$. As we mentioned above, the robot lacks perfect navigation skills and only holds low-level skills such as *"go forward"*, *"turn left/right"*, etc. Therefore, the robot needs to satisfy the constraint *"no obstacle in the front"*. If the constraint is violated, it must perform skill *"turn left/right"* to avoid the collision.

**(2) Move-box.** The robot is required to transport a certain box from one location to another. A proper solution might involve 1) Go to place A. 2) Pick up box. 3) Go to place B. 4) Put down box. We introduced additional perturbations to this task by assuming that the robot has a probability $p$ of dropping the box every second during transport.

**(3) Prepare-food.** The robot is required to collect 2-5 types of foods from random positions according to abstract language instructions (example in Figure 3b). We introduced additional perturbations to this task by assuming that the robot has a probability $p_1$ of failing to pick the object and $p$ of dropping the carried object every second. These tasks may need 10-20 steps of low-level skills.

### 5.2.2 VLM FINE-TUNING

In our experiments, we observed that the performance of zero-shot transferred VLM diminishes as the scene complexity increases, such as in the prepare-food task involving more than 20 objects. To address this, we collected a small dataset that only consisted of 5 demonstrations with 128 image-text pairs to fine-tune the BLIP-2 model (Li et al., 2023b). These 5 demonstrations only included fruit objects, while the test tasks involved entirely different scenarios, including unseen objects in random positions like junk food, vegetables, and seafood, as well as unseen backgrounds. It is worth noting that fine-tuning the VLM on the prepare-food task also yielded benefits for unseen tasks. We can use "detection time" to refer to the time interval between when the misalignment occurred and when detectors detected this violation. We find the fine-tuned VLM exhibited improved efficiency in detecting dropped boxes during move-box tasks, reducing the average "detection time" from 2.5 seconds to 0.6 seconds. Some out-of-distribution samples are shown in Figure 6. Ablations and analysis of the fine-tuned VLM can be found in Appendix B.4.

### 5.2.3 RESULTS

**Experiment details** Following the pipeline in Figure 2(c), we use Vicuna-13B (Chiang et al., 2023) as the LLM planner and BLIP-2 (Li et al., 2023b) as the VLM constraint detector. We use **DoReMi-FT** to denote DoReMi with VLM fine-tuned on the prepare-food task, as described in Sec. 5.2.2. We compare our methods with (1) **SayCan** (Ahn et al., 2022) which assumes every step is executed successfully, and (2) **Inner Monologue (IM)** (Huang et al., 2022b) which plans at the end of each step and uses the same vision-language model as both success detectors and scene descriptors. (3) **Periodic replan** which re-plans at a fixed time interval of 3 seconds and obtains feedback from the

| Tasks with disturbance | | SayCan | IM | Success Rate(%) ↑ Periodic replan | DoReMi (ours) | DoReMi-FT (ours) | IM-Oracle | Execution Time(s) ↓ DoReMi (ours) | DoReMi-FT (ours) | IM-Oracle |
|---|---|---|---|---|---|---|---|---|---|---|
| Obstacle-avoidance with density $d$ | $d$=0.0 | 100(±0) | 100(±0) | 100(±0) | 100(±0) | 100(±0) | 100(±0) | 24.2(±0.8) | 24.2(±0.8) | 24.2(±0.8) |
| | $d$=0.3 | 68(±6) | 68(±6) | 59(±8) | 92(±6) | 92(±6) | 68(±6) | 31.2(±2.4) | 31.2(±2.4) | - |
| | $d$=0.6 | 40(±8) | 40(±8) | 37(±10) | 90(±6) | 90(±6) | 40(±8) | 34.3(±3.2) | 34.3(±3.2) | - |
| Move-box with random drop $p$ | $p$=0.0 | 98(±2) | 98(±2) | 96(±3) | 97(±2) | 97(±2) | 98(±2) | 32.2(±2.5) | 32.2(±2.5) | 32.1(±2.5) |
| | $p$=0.02 | 61(±7) | 63(±7) | 55(±9) | 95(±4) | 96(±4) | 98(±2) | 38.4(±3.0) | 35.0(±3.4) | 46.5(±4.7) |
| | $p$=0.04 | 42(±9) | 46(±9) | 38(±8) | 94(±4) | 96(±4) | 96(±2) | 43.6(±3.5) | 37.3(±3.1) | 61.2(±7.6) |
| Prepare-food with pick failure $p_1$=0.1 random drop $p$ | $p$=0.0 | 78(±5) | 83(±4) | 81(±5) | 85(±6) | 96(±3) | 99(±1) | - | 27.6(±2.7) | 27.8(±3.0) |
| | $p$=0.02 | 49(±5) | 56(±5) | 50(±5) | 66(±4) | 93(±5) | 97(±2) | - | 31.0(±3.8) | 36.8(±5.8) |
| | $p$=0.04 | 18(±5) | 21(±7) | 16(±6) | 37(±8) | 91(±6) | 96(±2) | - | 35.2(±6.5) | 46.3(±7.5) |

Table 2: Success rates and task execution time under different degrees of disturbances. We only evaluate execution time under high task success rates. The results show the mean and standard deviation over 5 different seeds each with 20 episodes.

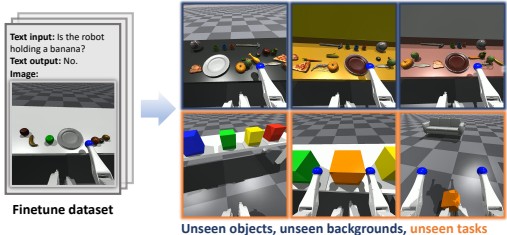

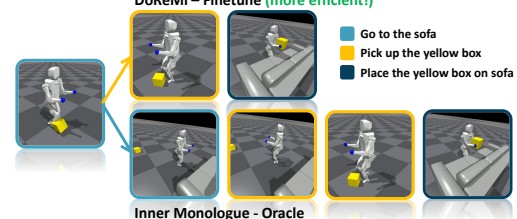

Figure 6: VLM detector fine-tuned on the small dataset can benefit unseen objects, unseen background, and unseen tasks.

Figure 7: Box dropped during the execution of skill "Go to the sofa". Inner Monologue only re-plans when the current skill is finished, taking more time to complete the task.

same VLM scene descriptors. (4) **IM-Oracle**. Inner monologue with Oracle feedback which does not exist in practical real-world settings.

**Result Analyses** The results are shown in Table 2. Similar to analysis in section 5.1, SayCan failed due to the absence of re-planning mechanisms and Inner-monologue failed because of the ambiguity and the low frequency of the feedback. Additionally, we find that naively increasing the re-plan frequency (Periodic replan baseline) does not necessarily improve success rates and can even lead to performance degradation. These results can be explained intuitively as follows: without sufficiently precise feedback, the more you re-plan, the more mistakes you may make. Higher frequency is beneficial only with precise enough feedback. These results further highlight the advance of DoReMi which enables more precise feedback, thanks to the seamless cooperation between LLMs and VLMs to propose and detect critical constraints.

In order to enhance the performance in extremely complex scenarios, such as the prepare-food task with over 20 objects, we fine-tuned the VLM on a small dataset as claimed in section B.4. DoReMi-FT with the fine-tuned BLIP-2 model performs better in all complicated scenes with unseen objects, unseen backgrounds, and even unseen tasks. For instance, in unseen move-box tasks, the detector can detect constraint violations more quickly and lead to a shorter total execution time. Furthermore, DoReMi-FT even surpasses IM-oracle in execution time while maintaining similar success rates due to its immediate detection and recovery mechanism, as depicted in Figure 7.

# 6 DISCUSSION

**Limitation** Our experiments indicate that the zero-shot transferred VLM is not a perfect constraint detector. We need to fine-tune the VLM in complicated tasks to improve detection accuracy and our framework can benefit from more advanced VLMs in the future. Furthermore, a detector fully based on vision may be limited by mis-detection, occlusion, and perspective. We may explore detectors in other modalities under our framework in the future.

**Conclusion** When employing language models for embodied tasks in a hierarchical approach, the low-level execution might deviate from the high-level plan. We emphasized the importance of continuously aligning the plan with execution and leveraged LLM to generate both plan and constraints, which enables grounding language through immediate detection and recovery. Theoretical analyses and a variety of challenging tasks in disturbed environments demonstrated the effectiveness of DoReMi.

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

# A PROOF FOR THEOREM

We denote the execution time of low-level skill with random variable $t$ with mean $\mathbb{E}[t] = \mu$ and variance $Var(t) = \sigma^2$. Misalignment can occur at any time $s$ within the execution time interval $[0, t]$ where $0 \leq s \leq t$. Additionally, we assume our constraint detector has probability $p_d$ to detect each misalignment. We define the discrete random variable $M$ as the number of misalignment occurrences under the following assumptions: (1) Plan-execution misalignments occur independently. (2) Misalignments occur at a constant ratio $\lambda$ within a small time interval: $\lim_{t \to 0} P(M = 1) = \lambda t$. (3) No two misalignments occur simultaneously: $\lim_{t \to 0} P(M = k) = 0$ for $k > 1$. Under these assumptions, the number of plan-execution misalignments follows a Poisson distribution (Papoulis & Unnikrishna Pillai, 2002):

$$P(M = k) = \frac{(\lambda t)^k e^{-\lambda t}}{k!} \quad k = 0, 1, 2, 3... \tag{4}$$

**Theorem 1** *The following equations describe the possible time-savings $t_s$ and the success rate improvement $P_s$ under immediate detection and re-planning:*

$$\mathbb{E}(t_s) = \sum_k P(M = k)\mathbb{E}(t_w|M = k) = \frac{p_d\lambda(\mu^2 + \sigma^2)}{2} - p_d\lambda\mu\Delta t \tag{5}$$

$$\mathbb{E}(P_s) = 1 - \mathbb{E}(e^{-\lambda t}) \approx p_d\lambda\mu - \frac{(2p_d - p_d^2)\lambda^2(\mu^2 + \sigma^2)}{2} \tag{6}$$

## A.1 LEMMA

**Lemma 1** *Given a Poisson process which is conditional on n arrivals in the time interval $(0, t)$, the conditional pdf (probability density function) of event occurrence time $t_1, t_2..., t_n$ satisfy (Papoulis & Unnikrishna Pillai, 2002):*

$$f(t_1, ..., t_n|M(t) = n) = \frac{n!}{t^n} \quad 0 \leq t_1 \leq ... \leq t_n \leq t \tag{7}$$

**Proof** Since the inter-arrival times of Poisson distribution are independent exponentially distributed, the joint pdf of the $n$ first arrival times is:

$$\begin{aligned} f(t_2, t_2, ..., t_n) &= f(t_1)f(t_2|t_1)...f(t_n|t_{n-1}) \\ &= \lambda e^{-\lambda t_1}\lambda e^{-\lambda(t_2-t_1)}...\lambda e^{-\lambda(t_n-t_{n-1})} \\ &= \lambda^n e^{-\lambda t_n} \end{aligned} \tag{8}$$

And conditional pdf can be derived:

$$\begin{aligned} f(t_1, t_2, ..., t_n|M(t) = n) &= \frac{f(t_2, t_2, ..., t_n, M(t) = n)}{P(M(t) = n)} \\ &= \frac{f(t_1, t_2, ..., t_n)P(M(t) = n|t_1, t_2, ..., t_n)}{P(M(t) = n)} \\ &= \frac{\lambda^n e^{-\lambda t_n} e^{-\lambda(t-t_n)}}{e^{-\lambda t}(\lambda t)^n/n!} \\ &= \frac{n!}{t^n} \end{aligned} \tag{9}$$

That is to say, each event can be considered as "placed" independently and uniformly at a given time in $[0, t]$.

## A.2 PROOF FOR THEOREM 1

**Soft Misalignment** Based on lemma A.1, each event can be considered to occur independently and uniformly at a given time in $[0, t]$, the event that occurs at $s$ will lead to time cost $t - s$. Total time cost without immediate replanning $(E(M) = \sum P(M = k) * k = \lambda t)$:

$$
\begin{aligned}
\mathbb{E}(t_{delay}) &= \sum_{t_i}(t - t_i) = \sum_k P(M = k)\mathbb{E}_t(t_w|M = k) \\
&= \mathbb{E}_t[\sum_k P(M = k) * kt/2] \\
&= \mathbb{E}_t[\lambda t^2/2] = \lambda(\mu^2 + \sigma^2)/2
\end{aligned}
\tag{10}
$$

Time cost without immediate replan (every event has detection time $\Delta t$ and failed detect $p_d$):

$$
\begin{aligned}
\mathbb{E}(t_{doremi}) &= (1 - p_d)\lambda(\mu^2 + \sigma^2)/2 + p_d\mathbb{E}_t[M] * \Delta t \\
&= (1 - p_d)\lambda(\mu^2 + \sigma^2)/2 + p_d\lambda\mu\Delta t
\end{aligned}
\tag{11}
$$

The wasted time $\mathbb{E}(t_w)$ is the difference between $\mathbb{E}(t_{delay})$ and $\mathbb{E}(t_{doremi})$:

$$
\mathbb{E}(t_w) = \mathbb{E}(t_{delay}) - \mathbb{E}(t_{doremi}) = \frac{p_d\lambda(\mu^2 + \sigma^2)}{2} - p_d\lambda\mu\Delta t
\tag{12}
$$

**Critical misalignment** Once critical misalignment comes, a delayed replanning will lead to failure. So the failure ratio $P_f$ equals the probability that the misalignment occurrence number is greater than 1. We assume misalignment happens ratio $\lambda$ is very small and we use second-order Tyler expansion to approximate the failure probability without immediate detection.

$$
\begin{aligned}
\mathbb{E}(P_f) &= \mathbb{E}_t[\sum_{k \geq 1} P(M = k)] = \mathbb{E}_t[1 - P(M = 0)] = 1 - \mathbb{E}_t(e^{-\lambda t}) \\
&= 1 - \mathbb{E}_t(1 - \lambda t + \lambda^2 t^2/2 + ...) \approx \lambda\mathbb{E}_t(t) - \lambda^2\mathbb{E}_t(t^2) \\
&= \lambda\mu - \frac{\lambda^2(\mu^2 + \sigma^2)}{2}
\end{aligned}
\tag{13}
$$

Since we consider the detector has probability $p_d$ to detect the violation and each violation happens independently, we can view the new process as Poisson distribution with $\lambda' = \lambda * (1 - p_d)$

$$
\begin{aligned}
\mathbb{E}(P_s) &= E(P_f) - E(P'_f) = \lambda\mu - \frac{\lambda^2(\mu^2 + \sigma^2)}{2} - ((1 - p_d)\lambda\mu - \frac{(1 - p_d)^2\lambda^2(\mu^2 + \sigma^2)}{2}) \\
&= p_d\lambda\mu - \frac{(2p_d - p_d^2)\lambda^2(\mu^2 + \sigma^2)}{2}
\end{aligned}
\tag{14}
$$

# B HUMANOID ROBOT TASK

## B.1 BASIC HUMANOID ROBOT INFORMATION

Our robot has 21 links and 20 degrees of joint freedom(DOF), and each joint holds a corresponding motor.

**Link names:** "base", "left shoulder pitch", "left shoulder roll", "left arm", "left elbow", "right shoulder pitch", "right shoulder roll", "right arm" "right elbow", "left leg yaw", "left leg roll", "left leg pitch", "left knee", "left ankle roll", "left ankle pitch" "right leg yaw", "right leg roll", "right leg pitch", "right knee", "right ankle roll", "right ankle pitch"

**DOF joint names:** "left shoulder pitch", "left shoulder roll", "left arm", "left elbow", "right shoulder pitch", "right shoulder roll", "right arm" "right elbow", "left leg yaw", "left leg roll", "left leg pitch", "left knee", "left ankle roll", "left ankle pitch" "right leg yaw", "right leg roll", "right leg pitch", "right knee", "right ankle roll", "right ankle pitch"

## B.2 Low-level Skill Training

Controlling complex humanoid robots with a single policy is challenging. Thus, we train low-level skills at the category level. Following the separate framework in (Ma et al., 2022), we utilize reinforcement learning to train locomotion policy and use model-based methods to obtain manipulation policy. In the case of our humanoid robot, there are 12 motors dedicated to the legs and 8 motors allocated to the arms. Notably, the observation of arm motors is not incorporated into the locomotion policies.

The locomotion policy is responsible for directly controlling the 12 motors associated with the legs, leading to a 12-dimensional action space. These policies output the target position of motors and run at 50 Hz, followed by PD controller run at 1000 Hz with $k_p = 100$ and $k_d = 2.5$. The proprioceptive observation space of the robot includes various dimensions: 12-dimensional joint angles, 12-dimensional joint angular velocities, 12-dimensional last actions, 3-dimensional angles between the torso and gravity, 2-dimensional periodic clock signals, and reserved 3-dimensional command signals, resulting in a total of 44 basic observation spaces.

We train low-level skills with the Deepmimic algorithm based on the Legged Gym (Isaac Gym Environments for Legged Robots) environment `https://github.com/leggedrobotics/legged_gym` built with the Isaac Sim physics simulator. Motion capture data we used can be found in the poselib `https://github.com/NVIDIA-Omniverse/IsaacGymEnvs/tree/main/isaacgymenvs/tasks/amp/poselib`.

**Locomotion Policy** This neural network policy is conditioned on 3-dimensional commands which respectively represent the required velocity in x-direction, y-direction, and required yaw angular velocity. In order to obtain natural moving gaits, we use the Deepmimic algorithm with multiple Motion Capture Date(Mocap data), thus the reward function has 2 parts including tracking commanded linear/angular velocity and imitating the style of Mocap data. This learned policy can help robots realize a category of sub-skills related to locomotion like: "Go forward fast", "Go forward at speed $v$", "Stand still", "Turn right/left", "Go to target place A", etc.

**Arm Manipulation Policy** Since we separate the control of arm and leg, we can use various manipulation policies including learned neural network policy or model-based policy, without influencing the leg locomotion policy. We use a linear interpolation controller to achieve the skill: "Pick up box", "Pick up box on the floor", "Put box on table", etc.

**Hyperparameters** Deepmimic algorithm pipeline is similar to PPO. Hyperparameters of the backbone Deepmimic algorithm can be found in table 3.

| Parameters | Value |
|---|---|
| Number of Environments | 4096 |
| Learning epochs | 5 |
| Steps per Environment | 24 |
| Minibatch Size | 24576 |
| Episode length | 20 seconds |
| Discount Factor | 0.99 |
| Generalised Advantage Estimation(GAE) | 0.95 |
| PPO clip | 0.2 |
| Entropy coefficient | 0.005 |
| Desired KL | 0.01 |
| Learning Rate | 5e-4 |
| Weight decay | 0.01 |

Table 3: Hyperparameters of backbone PPO algorithm.

**Training curves** The training process for the navigation policies and stand/squat policies is illustrated in Figure 8. The navigation policy enables the robot to control its xy position within the world frame, while the height switch policy allows for adjusting the robot's z height within the world frame.

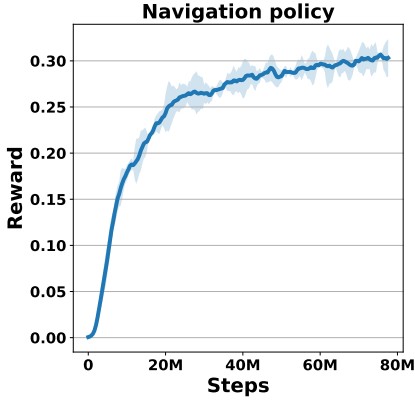

Figure 8: Traning curves for navigation policy by Deepmimic algorithm.

### B.3 TASK DETAILS

**Obstacle-avoidance task:** The finish line is 8m-16m far away from the robot. The task instruction is "Reach the finish line which is 15m in front of you". And the robot will plan low-level skills "go forward 15m". However, unknown obstacles may appear in the way and the robot needs to replan the skill "turn right/left" to avoid collision.

**Move-box task:** The robot is required to move the box in a certain color from place A to place B. The box color is randomly selected from {red, yellow, orange, green, blue, purple}.

**Prepare food:** The robot is required to collect 2-5 types of food. The food is in random positions. The robot has a chance to fail to pick up the food and the food may drop from the end-effector during transportation.

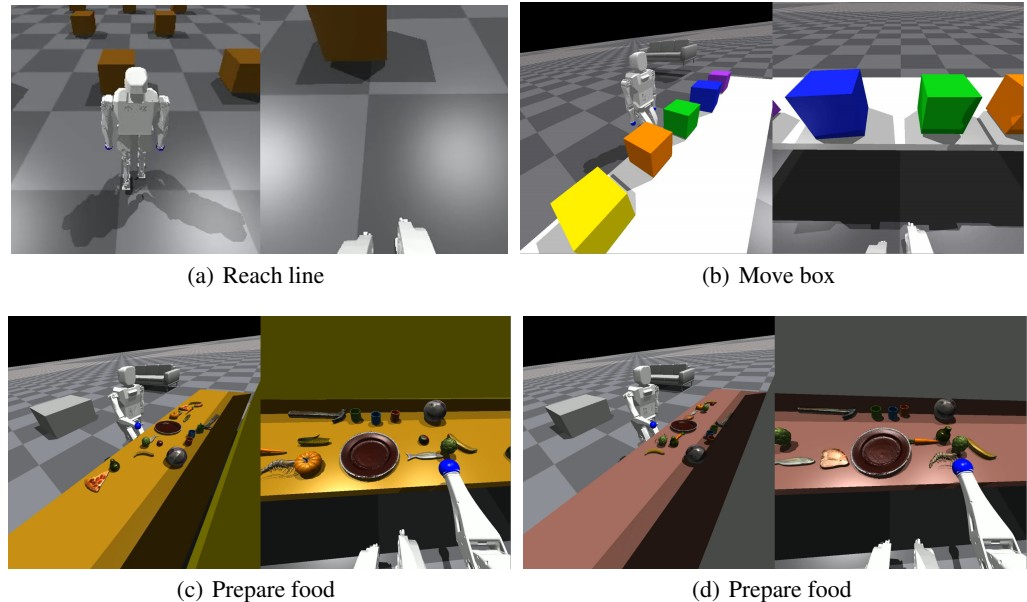

Figure 9: Robot and tasks. For every task, the photo on the Left is recorded from a third-view camera and the photo on the right is from the first-view camera. We use first-view images as inputs to VLM models.

### B.4 VISION LANGUAGE MODEL FINETUNING

Vision language model input images are the first-view camera attached to the robot. Visualization of this camera can be found in Figure 9. Zero-shot transferred BLIP-2 model performs well in the obstacle-avoidance task and the move-box task. However, in complex prepare-food tasks, the accuracy of the VLM detector drops significantly. To better handle complex tasks, we finetune BLIP-2 ViT-g FlanT5XL model (Li et al., 2023b) with 4.1B parameter on prepare food task, we use LoRA (Hu et al., 2021) method to finetune the parameters in an efficient war.

**Train dataset collection:** we collect 5 demonstration trajectories which are 128 seconds long. Then we label the data every 1 second, which results in a total of 128 image-text pairs. The demonstrations only contain fruit objects on a white plain table without decorations, as shown in Figure 10.

**Object in training dataset:** apple, green apple, strawberry, banana, orange, lemon.

**Unseen objects:** carrot, chicken, corn, fish, meat, peach, pear, pizza, pumpkin, shrimp, vegetable, soccer, cups, red plate, hammer.

**Unseen background:** The backgrounds are randomly generated colors, 3 samples are shown in Figure 10.

**Unseen tasks:** obstacle-avoidance and move-box are two unseen tasks. Compared to the zero-shot transferred BLIP-2 model, the finetuned BLIP-2 model performs better in move-box tasks and performs similarly in obstacle-avoidance tasks. A fine-tuned BLIP-2 model can discover box dropped more quickly.

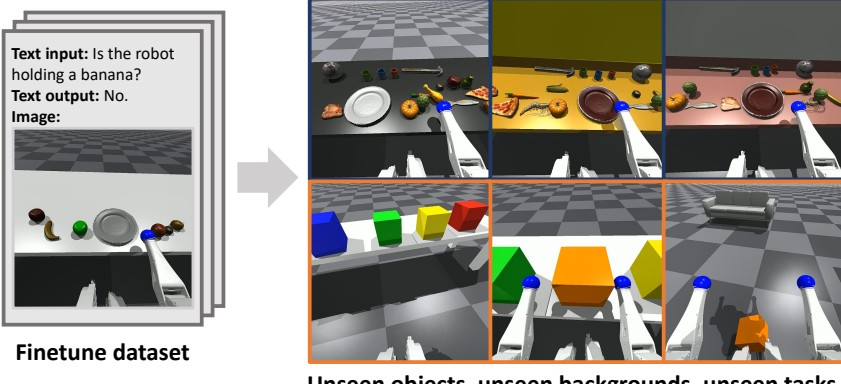

Figure 10: Train dataset only contains fruit objects on plain white tables. And the test tasks are much more complicated with unseen objects, backgrounds, and tasks.

**Ensemble VLM Answers:** The VLM text output for a single vision-question pair may not always be correct. To reduce the probability of true-negative(TN) samples, an ensemble approach can be utilized by incorporating 2 consecutive frames. Practically, We detect constraint violations by considering 2 consecutive time-step images where VLM identifies the same constraint violation. The time step duration, denoted as $\Delta t$, is set to 0.2 seconds.

**Fine-tuned VLM can benefit unseen objects and unseen backgrounds.** First, we compare the zero-shot transferred BLIP-2 model and fine-tuned BLIP-2 model in unseen objects and backgrounds. Zero-shot transferred BLIP-2 model may fail to detect a drop during transportation and lead to low task success rates. A fine-tuned BLIP-2 model can detect constraint violation with high accuracy and lead to high task success rates. Some comparisons are shown in Figure 11. we use a green border around the image to indicate that our VLM detector determines there is no constraint violation in the image, and a red border to indicate that VLM believes there is a constraint violation in the image.

**Fine-tuned VLM can benefit unseen tasks.** We also compare the zero-shot transferred BLIP-2 model and fine-tuned BLIP-2 model with unseen tasks. In move-box tasks, the zero-shot transferred BLIP-2 model can detect the box drop when the box is totally dropped on the floor or even disappears in the camera. However, the finetuned BLIP-2 model seems to figure out the correct robot body part

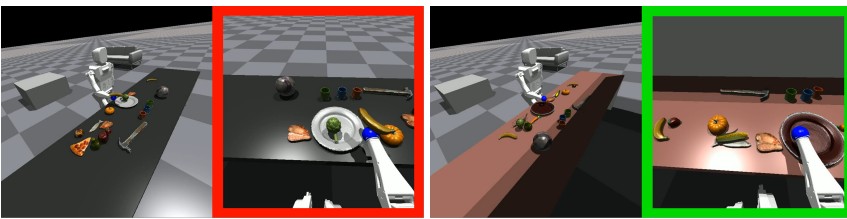

(a) Wrong detection of zero-shot transferred BLIP-2 model.

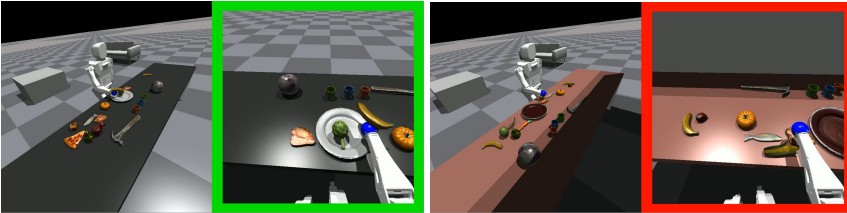

(b) Correct detection of finetuned BLIP-2 model on unseen objects and background

Figure 11: Fine-tuned VLM performs well in unseen objects and unseen backgrounds.

and can detect box drop as soon as the box leaves the robot arm, as shown in Figure 12. The average detection time decreased from 2.5 seconds to 0.4 seconds.

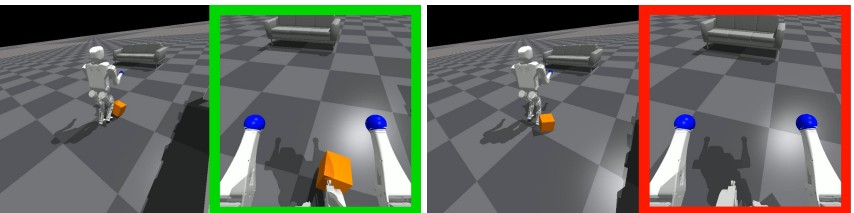

(a) Zero-shot transferred BLIP-2 identify box drop until box disappear in the horizon

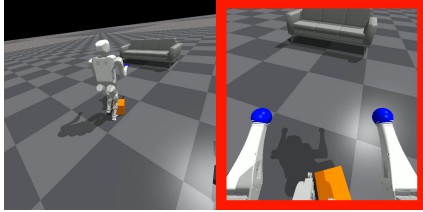

(b) Fine-tuned BLIP-2 immediately detect the box drop in this unseen task.

Figure 12: Fine-tuned VLM can benefit unseen tasks

**Accuracy analysis**

After fine-tuning, the accuracy of VLM in the prepare-food task has significantly increased, as shown in Table 4.

**VLM ablation study** We also conduct an ablation study on different types of VLM with max disturbances in our tasks, as shown in Table 5. The BLIP-2 model performs similarly to the Instruct-BLIP model. However, all zero-shot transferred models can not perform well in complicated prepare-food tasks.

| | Before finetune | | | | After finetune | | | |
|---|---|---|---|---|---|---|---|---|
| | TP | FN | FP | TN | TP | FN | FP | TN |
| Obstacle | 120 | 5 | 0 | 14 | 121 | 4 | 0 | 14 |
| Move box | 140 | 0 | 6 | 22 | 140 | 0 | 2 | 26 |
| Prepare food | 78 | 27 | 8 | 25 | 99 | 6 | 1 | 32 |

Table 4: Accuracy analysis of VLM on humanoid tasks.

| Success rate% | BLIP-1 | BLIP-2 | Instruct-BLIP |
|---|---|---|---|
| Obstacle-avoidance | 88 | 90 | 92 |
| Move-box | 64 | 94 | 92 |
| Prepare-food | 16 | 37 | 40 |

Table 5: Ablation study on zero-shot transferred VLM

## B.5 ABLATIONS IN REPLAN AND DETECTION TIME

**Re-plan time counts** The average numbers of planning in trajectories are shown in Table 6. We did not count the planning number in low success rate situations where agents may keep planning the wrong steps which are meaningless. It's worth mentioning that DoReMi only triggered a re-plan of LLM if the constraint detector identified a constraint violation. The replanning time of DoReMi is comparable to IM with Oracle feedback, indicating that our constraint detector provides very precise feedback and triggers replans at the correct condition.

| Number of planning | | Saycan | Inner-Monologue | Inner-Monologue Oracle | DoReMi-FT |
|---|---|---|---|---|---|
| | d=0.0 | 1.0 | 1.0 | 1.0 | 1.0 |
| Obstacle-avoidance | d=0.3 | 1.0 | 1.0 | 1.0 | 1.4 |
| | d=0.6 | 1.0 | 1.0 | 1.0 | 2.2 |
| | p=0.0 | 5.0 | 5.0 | 5.0 | 5.0 |
| Move-box | p=0.02 | - | - | 6.1 | 6.2 |
| | p=0.04 | - | - | 7.3 | 7.4 |
| | p=0.0 | 16.0 | 18.2 | 17.2 | 17.2 |
| Prepare-food | p=0.02 | - | - | 20.9 | 21.2 |
| | p=0.04 | - | - | 24.3 | 24.7 |

Table 6: Number of planning

**Ablations on constraint detection interval time** The ablation study on constraint detection interval was shown in Table 7. In obstacle tasks, the agent may not have enough time to change direction with too large constraint detection intervals. In other tasks, larger detection intervals resulted in longer execution times.

| success rate | 0.2s (original) | 0.4s | 0.6s | 1.0s | 1.5s |
|---|---|---|---|---|---|
| Obstacle | 90 | 90 | 89 | 83 | 74 |
| Move-box | 96 | 95 | 95 | 95 | 95 |
| Prepare-food | 91 | 89 | 90 | 90 | 87 |
| **Execution time** | | | | | |
| Obstacle | 34.3 | 34.6 | 34.2 | - | - |
| Move-box | 37.3 | 39.7 | 40.7 | 43.2 | 45.1 |
| Prepare-food | 35.2 | 36.8 | 38.0 | 40.3 | 43.8 |

Table 7: Ablation on detection interval times

## C ROBOT ARM MANIPULATION TASK

### C.1 IMPLEMENTATION DETAILS

**Low-level Policy** The low-level policy is similar to CLIPort(Shridhar et al., 2022) and Transporter Network (Zeng et al., 2020). Detailed references for its implementation can be found at `https://github.com/google-research/ravens`. This policy has been trained to perform single-step pick-and-place tasks based on language descriptions, and its performance is nearing perfection. However, for the purpose of our study, we presume this original policy to be perfect and introduce additional perturbations to the location placement.

**Ensembling Multi-step Detection by VLM** In the robot arm manipulation tasks, we use the zero-shot transferred BLIP-2 model. To decrease the true-negative error of the VLM detector, We detect constraint violations by considering 2 consecutive time-step images where VLM identifies the same constraint violation.

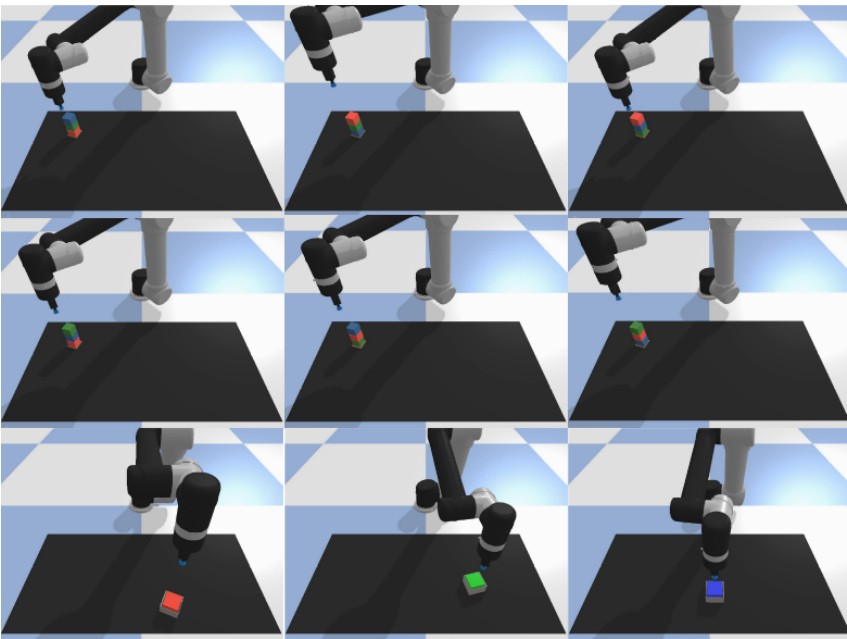

Figure 13: Our method is agnostic to different stack orders.

**Baseline** To adapt to our tasks, we slightly modify the original implementation of three baselines: (1) SayCan: similar to the original implementation of SayCan `https://github.com/google-research/google-research/tree/master/saycan`, we don't use a value function here given this environment does not have RL-trained policies or an associated value function. Instead, we use affordances obtained from ground-truth object information. The low-level

| Tasks with disturbance | | Success Rate(%) ↑ | | | Execution Time(s) ↓ | | |
|---|---|---|---|---|---|---|---|
| | | Inner Monologue | DoReMi (w/o ensembling) | DoReMi (ours) | Inner Monologue | DoReMi (w/o ensembling) | DoReMi (ours) |
| **Stack in order with noise $n$ random drop $p$=0.05** | $n$=0.0 | **100**(±**0**) | **100**(±**0**) | **100**(±**0**) | 7.9(±0.7) | 7.5(±0.5) | **7.4**(±**0.5**) |
| | $n$=1.0 | 94(±7) | 96(±4) | **98**(±**4**) | 9.3(±3.3) | 8.6(±2.9) | **8.1**(±**1.0**) |
| | $n$=2.0 | 83(±8) | 88(±7) | **94**(±**7**) | 17.3(±5.8) | 12.1(±2.9) | **10.8**(±**2.7**) |
| | $n$=3.0 | 63(±9) | 67(±10) | **73**(±**11**) | 36.3(±7.2) | 25.8(±7.1) | **19.9**(±**3.9**) |
| **Stack in order with noise $n$ random drop $p$=0.15** | $n$=0.0 | 92(±6) | 92(±6) | **94**(±**7**) | 10.6(±4.3) | 9.7(±3.2) | **8.9**(±**2.2**) |
| | $n$=1.0 | 88(±7) | 90(±7) | **92**(±**6**) | 14.8(±5.1) | 12.5(±4.2) | **10.3**(±**3.2**) |
| | $n$=2.0 | 73(±11) | 79(±9) | **85**(±**7**) | 25.2(±6.3) | 21.3(±5.7) | **14.0**(±**3.7**) |
| | $n$=3.0 | 23(±9) | 33(±10) | **44**(±**11**) | 47.8(±6.5) | 40.6(±6.9) | **29.3**(±**4.1**) |

Table 8: Ablation study over different degrees of perturbations and whether to adopt ensembling or not. The results show the mean and standard deviation over 4 different seeds, each with 12 episodes.

policies are adopted as the same as ours. We use Vicuna-13B (Chiang et al., 2023) to output the probabilities of each low-level policy in each scene. (2) CLIPort: the implementation is based on `https://github.com/google-research/ravens`. The oracle success detector is replaced with our VLM detector. (3) Inner Monologue: We reproduce the implementation based on (Huang et al., 2022b). Both the low-level policies and the LLM planner are the same as ours. The original success detector and the scene descriptor are also replaced with our VLM.

## C.2    Ablation Study

We evaluated the robustness of DoReMi in various environmental conditions by testing our method under distinct levels of perturbations. We observed that when the positional noise level $n$ of the end-effector exceeds 0.03 cm, the frequency of constraint violations escalates, leading to an almost zero success rate and dramatically increased execution time, despite accurate detection of constraint violations. This observation aligns with the theoretical expectation derived from a simple computation of block placement probabilities.

Our primary interest lies in the response of DoReMi to a spectrum of drop perturbation levels, in addition to the $p = 0.1$ presented in Table 1. As illustrated in Table 8, DoReMi demonstrates admirable performance under a variety of scenarios, outperforming the best-performing baseline. We attribute this largely to DoReMi's robust detection mechanism and its ability to swiftly recover from misalignment between plan and execution.

We are also curious to explore how the ensembling of multi-step detection would perform under various environmental settings. As indicated in Table 8, our findings suggest that ensembling can markedly enhance DoReMi's effectiveness in scenarios where strong perturbations exist and a single detection error could potentially result in a complete episode failure.

## C.3    Vision language model Analysis

**Accuracy analysis:** To analyze the accuracy of the VLM detector, we categorize all the detection results into True Positives (TP), True Negatives (TN), False Positives (FP), and False Negatives (FN), using these to calculate relevant accuracy metrics as outlined in Table 9.

A True Positive (TP) refers to the VLM correctly identifying that no constraint violation has occurred, whereas a True Negative (TN) signifies a successful detection of a constraint violation. A False Positive (FP) is when the VLM fails to recognize a constraint violation, and a False Negative (FN) is when the VLM incorrectly identifies a normal condition as a violation.

Utilizing the count in each of these categories, we compute the True Positive Rate (TPR) as $TPR = \frac{TP}{TP+FN}$. TPR reflects the accuracy with which the VLM identifies normal conditions. Similarly, the True Negative Rate (TNR) is calculated as $TNR = \frac{TN}{TN+FP}$, representing the accuracy of VLM in detecting constraint violations.

Further, we determine the Positive Prediction Value (PPV) as $PPV = \frac{TP}{TP+FP}$, and the Negative Prediction Value (NPV) as $NPV = \frac{TN}{TN+FN}$. These metrics correspond to the precision of the VLM detector in identifying normal conditions and constraint violations, respectively.

| Stack-block-in-order | | TP | TN | FP | FN | TPR | TNR | PPV | NPV |
|---|---|---|---|---|---|---|---|---|---|
| p=0 | n=0 | 150 | 0 | 0 | 0 | 1.00 | N/A | 1.00 | N/A |
| | n=1 | 188 | 0 | 0 | 4 | 0.98 | N/A | 1.00 | 0.00 |
| | n=2 | 249 | 27 | 3 | 6 | 0.98 | 0.90 | 0.99 | 0.82 |
| | n=3 | 272 | 81 | 1 | 2 | 0.99 | 0.99 | 1.00 | 0.98 |
| p=0.05 | n=0 | 173 | 5 | 0 | 0 | 1.00 | 1.00 | 1.00 | 1.00 |
| | n=1 | 204 | 7 | 1 | 1 | 1.00 | 0.88 | 1.00 | 0.88 |
| | n=2 | 196 | 23 | 3 | 4 | 0.98 | 0.88 | 0.98 | 0.85 |
| | n=3 | 253 | 92 | 2 | 6 | 0.98 | 0.98 | 0.99 | 0.94 |
| p=0.1 | n=0 | 202 | 13 | 1 | 1 | 1.00 | 0.93 | 1.00 | 0.93 |
| | n=1 | 180 | 8 | 0 | 1 | 0.99 | 1.00 | 1.00 | 0.89 |
| | n=2 | 212 | 14 | 3 | 1 | 1.00 | 0.82 | 0.99 | 0.93 |
| | n=3 | 231 | 100 | 4 | 7 | 0.97 | 0.96 | 0.98 | 0.93 |
| p=0.15 | n=0 | 182 | 24 | 1 | 0 | 1.00 | 0.96 | 0.99 | 1.00 |
| | n=1 | 178 | 12 | 2 | 0 | 1.00 | 0.86 | 0.97 | 1.00 |
| | n=2 | 175 | 26 | 1 | 3 | 0.98 | 0.96 | 0.99 | 0.90 |
| | n=3 | 208 | 128 | 6 | 9 | 0.96 | 0.96 | 0.97 | 0.93 |

Table 9: Statistics of VLM detection. The number of results of TP, TN, FP, FN are summed over 4 different seeds each with 12 episodes.

As per Table 9, both TPR and PPV maintain high values across various settings, which suggests that the VLM detector excels at identifying normal conditions. However, TNR is typically lower, particularly under conditions of low perturbations, indicating that the VLM detector may not be adept at detecting all constraint violations. The fluctuating detections become particularly pronounced when violations are infrequent. Similarly, NPV also trends lower across settings, signifying that our VLM detector might misidentify normal conditions as constraint violations at times, leading to redundant re-planning efforts.

## C.4   CASE VISUALIZATION

Similar to B.4, we use a green border around the image to indicate that our VLM detector determines there is no constraint violation in the image, and a red border to indicate that VLM believes there is a constraint violation in the image.

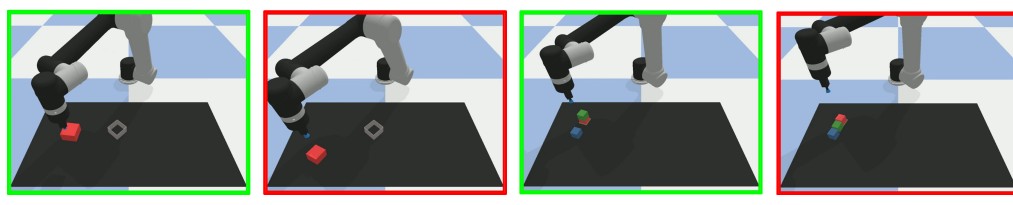

(a) **Q**:Is the robot holding the red block? **A**:Yes (b) **Q**:Is the robot holding the red block? **A**:No (c) **Q**:Is the green block on the red block? **A**:Yes (d) **Q**:Is the green block on the red block? **A**:No

Figure 14: Case visualization for robot arm experiment.

# D PROMPTS AND PIPELINE

We provide the prompt and the whole pipeline in this section. These prompts enable LLMs to generate both the next step and constraint through few-shot in-context learning. In practice, the $n^{th}$ generated constraint is used for $(n + 1)^{th}$ skill. We find this way is more natural for LLM and LLM can generate more admissible constraints.

## D.1 QUALITY OF LLM GENERATED CONSTRAINTS

Previous works demonstrate that LLM can finish high-level planning with high quality. Additionally, we conducted an experiment to analyze the quality of the constraints generated by LLMs.

**Constraint admissible rate.** We first conducted a user study to compare LLM-generated constraints and manually specified constraints. We sought the input of five individuals to assess the admissibility of these constraints, considering a constraint as admissible if at least four out of the five people reached a consensus. Our findings revealed the following: For 83 specific skills used in our tasks, LLM-generated constraints had a 98% admissible rate; For 50 random skills out of our tasks (like "open fridge" or "give milk to human"), they had a 94% admissible rate. These results underscore the remarkable proficiency of LLMs in generating constraints, driven by their profound understanding of the physical world. Some examples are shown in Table 10.

**Constraint consistent rate.** We then query the VLM with LLM-generated constraints and manually specified constraints under the same image input. The answers of the VLM under these two types of constraint inputs reach a consistency rate of **97%**, which proves that the LLM is able to generate very high-quality constraints and thus can be used for constraint generation.

| Specific Low-level Skill | LLM-generated Constraints | VQA Question |
|---|---|---|
| `go forward 10m` | +no obstacle in the front | +Is there any obstacle in the front? |
| `pick` block | +{**agent**}hold block | +Is the {**agent**} holding box? |
| `place` box on table | +box on table
-{**agent**}hold box | +Is the box on table?
-Is the {**agent**} holding box? |
| `open` fridge | +fridge is open | +Is fridge open? |
| `Give` milk to human | +human hold mild
-{**agent**}hold mild | +Is the human holding milk?
-Is the {**agent**} holding milk? |

Table 10: Examples of LLM-generated constraints. The symbol "**+**" indicates the addition of the constraint while "**-**" means popping out this constraint. Questions are in the general structure: "Is the {constraint}?"

## D.2 PROMPT

The robot performs manipulation tasks. At the same time, the robot needs to satisfy some constraints to ensure the successful execution of each task. Just fill in the blank and directly output the next step.

Task: Go forward

(0) Start, [Constraint: no obstacle in the front], (1) Go forward, [Constraint: no obstacle in the front], [Constraint violation: obstacle on the left], (2) Turn right, [Constraint: ],(3) Go forward, [Constraint: no obstacle in the front], (4) Done.

Task: Collect meat and banana on the plate.

(0) Start, [Constraint: ], (1) Go to meat, [Constraint: ], (2) Pick up meat, [Constraint: The robot is holding meat], (3) Go to plate, [Constraint: The robot is holding meat], [Constraint violation: The robot is not holding meat] (4) Pick up meat, [Constraint: The robot is holding meat], (5) Go to plate, [Constraint: The robot is holding meat], (6) Place meat in plate, [Constraint: ], (7) Go to banana, [Constraint: ], (8) Pick up banana, [Constraint: The robot is holding banana], (9) Go to plate, [Constraint: The robot is holding banana], (10) Place banana in plate, [Constraint: ], (10) Done.

Task: Stack blocks in the order of blue, red, and green.

(0) Start, [Constraint: ], (1) Pick the red block, [Constraint: The robot is holding red block], (2) Place the red block on the blue block, [Constraint: The red block is on the blue block], (3) Pick the green block, [Constraint: The robot is holding green block, the red block is on the blue block], (4) Place the green block on the red block, [Constraint: The red block is on the blue block, the green block is on the red block], [Constraint violation: the green block is not on the red block], (5) Pick the green block, [Constraint: The robot is holding green block, the red block is on the blue block], (6) Place the green block on the red block, [Constraint: The red block is on the blue block, the green block is on the red block], (7) Done.

## D.3 OVERALL PIPELINE

To help understand, here we provide a specific case in the timeline, the abstract task is "Stack blocks in the order of brown, red, and green."

1. **Planning and constraint generation for step 1:**

- LLM input: ...(prompt)...Task: Stack blocks in the order of brown, red, and green. (1)

- LLM output: Pick the red block, [Constraint: The robot is holding red block]

2. **Step 1 execution**

3. **Planning and constraint generation for step 2:**

- LLM input: ...(prompt)...Task: Stack blocks in the order of brown, red, and green.(1) Pick the red block, [Constraint: The robot is holding red block], (2)

- LLM output: Place the red block on the brown block, [Constraint: The red block is on the brown block]

4. **Step 2 execution**

5. **Planning and constraint generation for step 3:**

- LLM input: ...(prompt)...Task: Stack blocks in the order of brown, red, and green.(1) Pick the red block, [Constraint: The robot is holding red block], (2) Place the red block on the brown block, [Constraint: The red block is on the brown block], (3)

- LLM output: Pick the green block, [Constraint: The robot is holding green block, the red block is on the brown block]

6. **Step 3 execution**

7. **Planning and constraint generation for step 4:**

- LLM input: ...(prompt)...Task: Stack blocks in the order of brown, red, and green.(1) Pick the red block, [Constraint: The robot is holding red block], (2) Place the red block on the brown block, [Constraint: The red block is on the brown block], (3) Pick the green block, [Constraint: The robot is holding green block, the red block is on the brown block], (4)

- LLM output: Place the green block on the red block, [Constraint: The red block is on the brown block, the green block is on the red block]

8. **Step 4 execution, Detector: the red block is not on the brown block**

9. **Planning and constraint generation for step 5:**

- LLM input: ...(prompt)...Task: Stack blocks in the order of brown, red, and green.(1) Pick the red block, [Constraint: The robot is holding red block], (2) Place the red block on the brown block, [Constraint: The red block is on the brown block], (3) Pick the green block, [Constraint: The robot is holding green block, the red block is on the brown block], (4) Place the green block on the red block, [Constraint: The red block is on the brown block, the green block is on the red block], [**Detector**: the red block is not on the brown block], (5)

- LLM output: Pick the red block, [Constraint: The robot is holding red block]

10. **Step 5 execution**

11. **Planning and constraint generation for step 6:**

- LLM input: ...(prompt)...Task: Stack blocks in the order of brown, red, and green.(1) Pick the red block, [Constraint: The robot is holding red block], (2) Place the red block on the brown block, [Constraint: The red block is on the brown block], (3) Pick the green block, [Constraint: The robot is holding green block, the red block is on the brown block], (4) Place the green block on the red block, [Constraint: The red block is on the brown block, the green block is on the red block], [Detector: the red block is not on the brown block], (5) Pick the red block, [Constraint: The robot is holding red block], (6)

- LLM output: Place the red block on the brown block, [Constraint: The red block is on the brown block]

12. **Step 6 execution**

13. **Planning and constraint generation for step 7:**

- LLM input: ...(prompt)...Task: Stack blocks in the order of brown, red, and green.(1) Pick the red block, [Constraint: The robot is holding red block], (2) Place the red block on the brown block, [Constraint: The red block is on the brown block], (3) Pick the green block, [Constraint: The robot is holding green block, the red block is on the brown block], (4) Place the green block on the red block, [Constraint: The red block is on the brown block, the green block is on the red block], [Detector: the red block is not on the brown block], (5) Pick the green block, [Constraint: The robot is holding green block, the red block is on the brown block], (6) Place the red block on the brown block, [Constraint: The red block is on the brown block], (7)

- LLM output: Pick the green block, [Constraint: The robot is holding green block, The red block is on the brown block]

14. **Step 7 execution**

15. **Planning and constraint generation for step 8:**

- LLM input: ...(prompt)...Task: Stack blocks in the order of brown, red, and green.(1) Pick the red block, [Constraint: The robot is holding red block], (2) Place the red block on the brown block, [Constraint: The red block is on the brown block], (3) Pick the green block, [Constraint: The robot is holding green block, the red block is on the brown block], (4) Place the green block on the red block, [Constraint: The red block is on the brown block, the green block is on the red block], [Detector: the red block is not on the brown block], (5) Pick the green block, [Constraint: The robot is holding green block, the red block is on the brown block], (6) Place the red block on the brown block, [Constraint: The red block is on the brown block], (7) Pick the green block, [Constraint: The robot is holding green block, The red block is on the brown block], (8)

- LLM output: Place the green block on the red block, [Constraint: The red block is on the brown block, The green block is on the red block]

16. **Step 8 execution**

17. **Planning and constraint generation for step 9:**

- LLM input: ...(prompt)...Task: Stack blocks in the order of brown, red, and green.(1) Pick the red block, [Constraint: The robot is holding red block], (2) Place the red block on the brown block, [Constraint: The red block is on the brown block], (3) Pick the green block, [Constraint: The robot is holding green block, the red block is on the brown block], (4) Place the green block on the red block, [Constraint: The red block is on the brown block, the green block is on the red block], [Detector: the red block is not on the brown block], (5) Pick the green block, [Constraint: The robot is holding green block, the red block is on the brown block], (6) Place the red block on the brown block, [Constraint: The red block is on the brown block], (7) Pick the green block, [Constraint: The robot is holding green block, The red block is on the brown block], (8) Place the green block on the red block, [Constraint: The red block is on the brown block, The green block is on the red block]

- LLM output: Done.

18. **Step 9 execution**

