# OpenReview forum: "DoReMi: Grounding Language Model by Detecting and Recovering from Plan-Execution Misalignment"
_ICLR.cc/2024/Conference — Submitted to ICLR 2024_

### Official Review · Reviewer_WGrL · 2023-10-22

**Soundness:** 3 good
**Presentation:** 2 fair
**Contribution:** 3 good
**Rating:** 6
**Confidence:** 5

**Summary:**

This paper proposes a framework that empowers LLM-based planners with online failure detection and re-planning capabilities for robot tasks. The authors recruit a VLM (fine-tuned optionally) to continually monitor the satisfaction of a set of constraints (generated by an LLM), and identify a failure of action execution when any constraints are violated. Upon detecting a failure, the current action's low-level execution is halted and the LLM-based planner is invoked for re-planning. The proposed method is assessed in both a manipulation environment and a humanoid robot setting compared against various baselines. Quantitative results show that it has a higher success rate thanks to the accurate feedback from the VLM, and less execution time due to the immediate failure detection mechanism.

**Strengths:**

The paper presents a practical solution to grounding LLM-based planners for failure-aware re-planning. It proposes a novel way to acquire environmental feedback: it leverages an LLM to generate constraints, which are then examined by a VLM. This form of feedback is more precise than the open-ended scene captioning.

The benefits of the framework are sufficiently justified with comprehensive quantitative experiments.

**Weaknesses:**

The primary weakness of this paper lies in its presentation, which lacks clarity and hampers the comprehensibility of several sections, especially Section 3-5. I believe that the paper would greatly benefit from additional work on its writing and organization before it can be considered for publication.

My suggestions are as follows:

1. Remove redundant and repeated paragraphs.
    - Sentences introducing previous work in “LLM-based planning with feedback” appear three times in Section 2 (related works), Section 3 (problem statement), and Section 4.1 (method) respectively.
    - In experiments, most of the baselines are shared in the two environments, but explained twice in Section 5.1 and 5.2.
    - The first sentence in the abstract and introduction and very much similar. It would be nice to rewrite either one of them.
2. Redistribute the content in the main body and appendix, to avoid having terms and numbers declared in the main sections but never explained until in the Appendix.
    - In Section 4.2: “LLM-generated constraints reach an admissible rate of 98%” and “these two answers reach a consistent rate of 97%”
    - In Section 4.4: how are the “potential wasted time” and “failure probability” defined?
    - In Section 5.2.2: “the average detection time from 2.5 seconds to 0.6 seconds”, what is the average detection time?
3. Please review the paper's grammar, including verb tense, singular/plural form, and article usage. A few examples in Section 3:
    - assist LLM generate → assist LLMs to generate
    - such oracle feedback is → such oracle feedback are
    - the humans → humans
    - which leverage powerful LLM → which leverage powerful LLMs
    - high-level plan → high-level plans

**Questions:**

1. I have doubts in the usage of the Saycan baseline. It seems that Saycan generates one action per step by jointly considering both visual affordance and LLM output, rather than “decomposing a task into actions and assumeing execution success”. It might be more appropriate to use an alternative name for this baseline.
2. In the “Accuracy analysis” section on page 18 of the Appendix, I noticed that the sum of TP and TN are the same before and after fine-tuning the VLM. Does it indicate that there is no performance improvement after the fine-tuning?

---

> ### Author Response · Authors · 2023-11-13
> **Response to Reviewer WGrL**
>
> We are deeply grateful for your time and effort in reviewing our paper. Your meticulous reading and proofreading efforts are truly appreciated. We hope that the updated PDF and our forthcoming response will successfully address and resolve all of your concerns.
>
> ---
>
> **Q1: The primary weakness of this paper lies in its presentation, I believe that the paper would greatly benefit from additional work on its writing and organization**
>
> **Response:** Thank you for your thorough and meticulous feedback. Following your suggestions, we have implemented several changes to the manuscript. To aid in your review, all modifications in the updated document are highlighted **in blue text** for easy identification.
>
> - Refined the first sentence of the introduction to distinguish it from the abstract.
> - Remove the repeated mentions of “LLM-based planning with feedback” in Sections 3 and 4.1.
> - Move explanations of "admissible rate" and "consistency rate" from the appendix to the main paper in Section 4.2.
> - Change “potential wasted time” and “failure probability” to "time-saving" and "success rate improvement" which had appeared earlier in the paper
> - Add explanation on "detection time" in section 5.2.2
> - Simplified the repeated baseline analyses in Section 5.2. We also add explanations to fine-tuning effect and a new baseline which did not appear in Section 5.1.
> - The modified PDF corrected all the grammar mistakes you pointed out. (Except for the "feedbacks", the wiki dictionary said it is an uncountable noun, so we keep the "feedback"). Additionally, three co-authors double-checked all grammar carefully, and we also used GPT-4 and Grammarly to refine the grammar further.
>
> We hope all the modifications above can improve our presentation, and thanks for your constructive suggestions!
>
> ---
>
> **Q2: I have doubts in the usage of the Saycan baseline. It seems that Saycan generates one action per step by jointly considering both visual affordance and LLM output, rather than “decomposing a task into actions and assuming execution success”.**
>
> **Response:**  Sorry for the confusion. Our usage of Saycan essentially follows the usage in the Inner-monologue paper [1] (the follow-up work of the original SayCan authors). On page 3, lines 2-5 of [1], it is claimed that "SayCan assumes that each proposed step is executed successfully by the agent", and we made similar claims as the original authors. This is because SayCan adds every executed step to the prompt; thus, the LLM is unaware of failures, and value functions are difficult to reflect changes in environments precisely.
>
> Moreover, value functions only exist in the reinforcement learning low-level policies and are absent in other policy types, such as imitation and model-based policies. On page 11, section 6 of the SayCan paper [2], they instead adopt an object detector as the value function, which we also implemented in our usage. We hope these detailed explanations can address your concern.
>
> [1] Huang W, Xia F, Xiao T, et al. Inner monologue: Embodied reasoning through planning with language models[J]. arXiv preprint arXiv:2207.05608, 2022.
>
> [2] Ahn M, Brohan A, Brown N, et al. Do as i can, not as i say: Grounding language in robotic affordances[J]. arXiv preprint arXiv:2204.01691, 2022.
>
> ---
>
> **Q3: In the “Accuracy analysis” section on page 18 of the Appendix, I noticed that the sum of TP and TN are the same before and after fine-tuning the VLM. Does it indicate that there is no performance improvement after the fine-tuning?**
>
> **Response:**  We apologize for the typos in the table column names. The original table had a mix-up in the order of "FN" and "TN". The corrected table is as follows:
>
> | **Before fine-tuning** | **TP**  | FN   | FP   | **TN** | **After fine-tuning** | **TP**  | FN   | FP   | **TN** |
> | ---------------------- | ------- | ---- | ---- | ------ | --------------------- | ------- | ---- | ---- | ------ |
> | Obstacle               | **120** | 5    | 0    | **14** | Obstacle              | **121** | 4    | 0    | **14** |
> | Move box               | **140** | 0    | 6    | **22** | Move box              | **140** | 0    | 2    | **26** |
> | prepare food           | **78**  | 27   | 8    | **25** | prepare food          | **99**  | 6    | 1    | **32** |
>
> The results of "TP+TN" remain similar in the Obstacle and Move-box tasks before and after VLM fine-tuning, maintaining high accuracy. However, for the complex prepare-food tasks, fine-tuning with a small dataset significantly improved performance. The accuracy of the VLM detectors also aligns with the success rates of DoReMi and DoReMi-FT in Table 2.
>
> Thank you for carefully reviewing our papers, which helped us improve the paper. We hope our response above can solve your concerns and show the improved quality of our paper. If you have any further concerns, please do not hesitate to ask; we are always ready to answer!

---

> > ### Comment · Reviewer_WGrL · 2023-11-19
> >
> > I would like to thank the authors for their time and efforts to revise the paper and thoroughly respond to my concerns. After careful consideration, I am pleased to inform you that I have adjusted my rating from 5 to 6.

---

> > > ### Author Response · Authors · 2023-11-19
> > > **Thank you for your endorsement and the increased score!**
> > >
> > > Dear Reviewer WGrL:
> > >
> > > We are delighted to receive your response! We are profoundly grateful for your endorsement of our paper!
> > >
> > > Your support is immensely encouraging, and we are committed to diligently refining and further enhancing our manuscript to meet the highest standards.
> > >
> > > With best regards,
> > > The Authors

---

> ### Comment · Area_Chair_fsji · 2023-11-19
>
> Thanks for your help reviewing the paper!
>
> The authors provide a response to your review several days ago. If you haven't already, please be sure to read the other reviews as well as the authors' responses, and share any additional comments that you have with the authors and other reviewers.

---

> ### Author Response · Authors · 2023-11-19
> **Reply to Area Chair fsji and Reviewer WGrL (Official Comment by Authors)**
>
> Dear Area Chair fsji and Reviewer WGrL:
>
> We show our deepest thanks to AC for facilitating our discussion and thanks to Reviewer WGrL again for reviewing our paper!
>
> Following the suggestion of Reviewer WGrL, we have meticulously enhanced our presentations to clarify and enhance the quality of our paper. We hope these improvements have adequately addressed your concerns and effectively showcased the presentation of our paper.
>
> Last but not least, we remain open and prepared to make any additional modifications to our presentation as may be necessary. Please feel free to reach out with any additional questions; we are always available and willing to address any further inquiries you may have!
>
> Best wishes,
>
> Authors

---

### Official Review · Reviewer_nVnU · 2023-10-29

**Soundness:** 3 good
**Presentation:** 3 good
**Contribution:** 3 good
**Rating:** 6
**Confidence:** 4

**Summary:**

This paper proposed DoReMi, which enables frequent feedback into LLM planning. DoReMi uses LLM in a dual role of generating high-level plan as well as constraints for low-level skill execution; the constraints are then checked during skill execution by a second VLM. Detection of execution failure is formatted into a prompt for the LLM to facilitate re-planning. DoReMi

**Strengths:**

This paper is adequately original and significant. It introduces a VLM-based constraint detection to speed up LLM planning and recovery for robotics tasks. While prior work (Du et al., 2023) has considered using VLM for success detection, DoReMi automates the constraint criteria via common sense reasoning of LLM. I think this is a creative and useful combination of LLM and VLM together in a robust framework for task-and-motion planning. Given that large foundation models are increasingly deployed in decision-making, this work provides a conceptually appealing way to improve the robustness and safety of such integrated planning pipelines.

The paper is well-written, clear, and contains many useful diagrams and algorithm blocks that help convey the main messages.

**Weaknesses:**

There are several weaknesses in the paper, mostly lying in the strengths of the experimental results.

1. IM-Oracle seems to outperform DoReMi in most cases in terms of Success Rates at the cost of a slightly higher Execution Time(s). This makes me wonder whether DoReMi's improvement comes just from the ability to prematurely stops low-level skill execution. This ability seems quite easy to add to any other method as well.

2. The method appears to work well for low-level skills that immediately induce a state change in the interacted object (e.g., picking up an object); these skills are also precisely the ones that can be detected via Yes-No VQA queries. Relatedly, when the task becomes more complicated, DoReMi requires fine-tuning of the VLM to work well.

**Questions:**

1. Is the only difference between DoReMi and Inner-Monologue (Oracle) that DoReMi is able to utilize the VLM constraint feedback during skill execution while IM (Oracle) waits until the end of skill execution?

2. Could the number of re-planning be reported in the results section? Relatedly, I wonder how would changing the constraint checking time impact the performance of DoReMi?

3. A baseline that probabilistically calls for re-planning at fixed time interval should be included. This will help understand whether the advantages of DoReMi come from the ability of mid-execution replanning or the fact that the VLM constraint detection is accurately informing the replanning.

---

> ### Author Response · Authors · 2023-11-13
> **Response to Reviewer nVnU (Part 1/3)**
>
> We sincerely thank you for the time and effort you have invested in reviewing our paper and are truly impressed by your thorough understanding of our work! We hope that the updated PDF and our subsequent responses will adequately address your concerns.
>
> ---
>
> > **Q1: IM-Oracle seems to outperform DoReMi in most cases in terms of Success Rates at the cost of a slightly higher Execution Time(s). This makes me wonder whether DoReMi's improvement comes just from the ability to prematurely stops low-level skill execution. This ability seems quite easy to add to any other method as well.**
>
> **Response:** We sincerely thank you for this insightful comment. First of all, to address your concern, the enhancement offered by DoReMi stems from two key areas: **1. More precise feedback** and **2. Feedback at a higher frequency**. The enhanced precision, surpassing that of traditional scene descriptions, is achieved by guiding VLMs to focus on constraints generated by LLMs, as illustrated in Figure 3. Furthermore, our VLM selects binary answers, which is time efficient and triggers an immediate replanning process if constraints are violated, a functionality that was challenging to achieve in previous frameworks. In the subsequent ablation study presented in Q5's response, we further demonstrate the critical role of both precision and frequency in enabling timely recovery.
>
> Secondly, we would like to emphasize that comparing DoReMi directly with IM-Oracle is unfair. IM-Oracle operates under the assumption of having access to perfect feedback, a condition that is often unattainable in real-world scenarios. In contrast, DoReMi obtains feedback from a practical VLM informed by LLM, which is significantly more accurate than passive scene captioning. The inclusion of IM-Oracle in our analysis illustrates the theoretical upper bound of the IM framework's performance. We are encouraged to find that by incorporating LLM reasoning, practical VLMs (with optional fine-tuning) can provide highly precise feedback for our tasks, achieving success rates comparable to those of IM-Oracle.
>
>
>
> ---
>
> > **Q2: The method appears to work well for low-level skills that immediately induce a state change in the interacted object (e.g., picking up an object); these skills are also precisely the ones that can be detected via Yes-No VQA queries. Relatedly, when the task becomes more complicated, DoReMi requires fine-tuning of the VLM to work well.**
>
> **Response:** We deeply value your profound insight. In our experiments, we meticulously considered a broad range of commonly utilized robot skill sets and various disturbances that existed in different environments and robotic systems. These skill sets encompass prevalent tasks such as object picking, placement, and locomotion. The disturbances we considered are multifaceted, occurring randomly, including failure in picking, random drops during transportation, noise in placement position, unexpected obstacles on the way, etc.
>
> We totally agree complex situations exist that are challenging to detect by VLMs. In our experiments, we need to fine-tune the VLM to improve detection accuracy in complex situations, and our framework can benefit from more advanced VLMs in the future. Furthermore, as described in the limitation section (Sec. 6), our framework is also compatible with detectors in other modalities such as force, tactile, and audio. Therefore, we believe our framework is a general and scalable framework that can potentially solve more complex situations.
>
> ---
>
> > **Q3: Is the only difference between DoReMi and Inner-Monologue (Oracle) that DoReMi is able to utilize the VLM constraint feedback during skill execution while IM (Oracle) waits until the end of skill execution?**
>
> **Response:**  As discussed in Q1, the difference in feedback timing is just one aspect.  Another significant advantage of DoReMi is its more precise feedback. The original Inner-Monologue study considered a success detector, passive scene descriptor, and human feedback, which are either manually designed or challenging to obtain with high frequency. In contrast, DoReMi can achieve more accurate and frequent feedback by leveraging LLMs to reason about constraints.

---

> ### Author Response · Authors · 2023-11-13
> **Response to Reviewer nVnU (Part 2/3)**
>
> > **Q4: Could the number of re-planning be reported in the results section? Relatedly, I wonder how would changing the constraint checking time impact the performance of DoReMi?**
>
> **Response:** We conducted additional experiments as you suggested. However, due to the nine-page limit, we cannot include all these experiments in the main paper. We have added them to Appendix B.5 of the updated manuscript.
>
> The average numbers of planning in trajectories are as follows. We did not count the planning number in low success rate situations where agents may keep planning the wrong steps, which are meaningless. It's worth mentioning that DoReMi only triggers a re-plan of LLM if the constraint detector identifies a constraint violation.  The re-planning time of DoReMi is comparable to IM with Oracle feedback, indicating that our constraint detector provides very precise feedback and triggers re-planning correctly and timely.
>
>
> | Numbers of plan    |        | saycan |  IM  | IM-Oracle | DoReMi-FT |
> | ------------------ | ------ | ------ | :--: | --------- | --------- |
> | obstacle-avoidance | d=0.0  | 1.0    | 1.0  | 1.0       | 1.0       |
> |                    | d=0.3  | 1.0    | 1.0  | 1.0       | 1.4       |
> |                    | d=0.6  | 1.0    | 1.0  | 1.0       | 2.2       |
> | Move-box           | p=0.0  | 5.0    | 5.0  | 5.0       | 5.0       |
> |                    | p=0.02 | -      |  -   | 6.1       | 6.2       |
> |                    | p=0.04 | -      |  -   | 7.3       | 7.4       |
> | Prepare-food       | p=0.0  | 16.0   | 18.2 | 17.2      | 17.2      |
> |                    | p=0.02 | -      |  -   | 20.9      | 21.2      |
> |                    | p=0.04 | -      |  -   | 24.3      | 24.7      |
>
>
>
> The ablation study on constraint detection interval is shown as follows. In obstacle tasks, the agent may not have enough time to change direction with too large constraint detection intervals, leading to lower success rates. In other tasks, larger detection intervals result in longer execution times.
>
> | success rate       | 0.2s (original) | 0.4s | 0.6s | 1.0s | 1.5s |
> | ------------------ | --------------- | ---- | ---- | ---- | ---- |
> | Obstacle           | 90              | 90   | 89   | 83   | 74   |
> | Move-box           | 96              | 95   | 95   | 95   | 95   |
> | Prepare-food       | 91              | 89   | 90   | 90   | 87   |
> | **Execution time** |                 |      |      |      |      |
> | Obstacle           | 34.3            | 34.6 | 34.2 | -    | -    |
> | Move-box           | 37.3            | 39.7 | 40.7 | 43.2 | 45.1 |
> | Prepare-food       | 35.2            | 36.8 | 38.0 | 40.3 | 43.8 |

---

> ### Author Response · Authors · 2023-11-13
> **Response to Reviewer nVnU (Part 3/3)**
>
> >**Q5: A baseline that probabilistically calls for re-planning at fixed time intervals should be included. This will help understand whether the advantages of DoReMi come from the ability of mid-execution replanning or the fact that the VLM constraint detection accurately informs the replanning.**
>
> **Response:**  We really appreciate your insightful comments! They inspire us a lot. As we mentioned above, the advance of DoReMi comes from both precision and high frequency. We have made a table to better present these 2 aspects, with columns representing the quality of feedback and rows representing the feedback frequency. This allows us to better compare with the baselines:
>
> |                                 | (Relatively) worse feedback                   | (Relatively) preciser feedback |
> | ------------------------------- | --------------------------------------------- | ------------------------------ |
> | **(Relatively) low frequency**  | Inner monologue                               | Inner monologue - Oracle       |
> | **(Relatively) high frequency** | **periodic re-plan (baseline you suggested)** | DoReMi(-FT)                    |
>
> In the original manuscript, we have covered three blocks of this table, and the baseline you suggested fills the last "missing piece". Thank you for your insightful contribution! We present the experimental results below: (The complete results have been added to Table 2 in the revised manuscript.)
>
> | success rate(%)    |        | IM   | *periodic re-plan* | IM-Oracle | DoReMi-FT |
> | ------------------ | ------ | ---- | :----------------: | --------- | --------- |
> | obstacle-avoidance | d=0.0  | 100  |        100         | **100**   | **100**   |
> |                    | d=0.3  | 68   |         59         | 68        | **92**    |
> |                    | d=0.6  | 40   |         37         | 40        | **90**    |
> | Move-box           | p=0.0  | 98   |         96         | **98**    | **97**    |
> |                    | p=0.02 | 63   |         55         | **98**    | **96**    |
> |                    | p=0.04 | 46   |         38         | **96**    | **96**    |
> | Prepare-food       | p=0.0  | 83   |         81         | **99**    | **96**    |
> |                    | p=0.02 | 56   |         50         | **97**    | **93**    |
> |                    | p=0.04 | 21   |         16         | **96**    | 91        |
>
> We notice that directly increasing the re-planning frequency does not necessarily improve success rates and can even lead to performance degradation.  This result can be explained intuitively as follows:  **without sufficiently precise feedback, the more you re-plan, the more mistakes you may make. Higher frequency is beneficial only with precise enough feedback.** These results further highlight the advance of DoReMi which enables more precise feedback, thanks to the seamless cooperation between LLMs and VLMs to propose and detect critical constraints.
>
> ---
>
> We sincerely appreciate your time and efforts in reviewing our paper again! We hope our response above can solve your concerns and show the improved quality of our paper. If you have any further concerns, please do not hesitate to ask; we are always ready to answer!

---

> ### Author Response · Authors · 2023-11-19
> **Dear Reviewer nVnU:**
>
> Dear Reviewer nVnU:
>
> We are sincerely grateful for the time and effort you have invested in reviewing our paper, and we deeply appreciate your insightful feedback.
>
> Based on your comments, we conducted additional experiments to verify the effectiveness of our method. We hope these modifications could address your concern and show the improved quality of our paper.
>
> Furthermore, we are always available and willing to address any further inquiries you may have! Please don't hesitate to provide any viewpoints on our manuscript.
>
> With best regards,
>
> The Authors

---

> ### Author Response · Authors · 2023-11-22
> **Dear Reviewer nVnU: Thanks for your valuable time and efforts in reviewing our paper!**
>
> Dear Reviewer nVnU,
>
> Thanks for your valuable time and efforts in reviewing our paper! We understand that you may be rather busy during this discussion period. As the rebuttal stage is drawing to a close within a day, we would like to kindly request your feedback on our responses. We are happy to discuss with you in detail if you have additional comments about our paper. :)
>
> Look forward to your reply.
>
> With deepest gratitude,
>
> The Authors

---

> > ### Comment · Reviewer_nVnU · 2023-11-22
> >
> > Dear Authors,
> >
> > Thank you for your responses! I have elected to keep my current score given that it is already high. While some of my concerns are being addressed, I do agree with another reviewer in that the proposed method appears to be some system components (i.e., what kind of success feedback is provided, frequency of re-planning) that are not novel methodologically in themselves and can be implemented for other baselines. For example, IM-Oracle appears to do worse because it uses a less flexible form of feedback, but I do not think that those specific forms of feedback are core to what constitutes the "Inner Monologue" method; would IM combined with the type of (fine-tuned) VLM feedback provided to Doremi work well?

---

> ### Author Response · Authors · 2023-11-22
> **Dear Reviewer nVnU: We are here to answer your questions!**
>
> We are delighted to receive messages from you and have the chance to make further explanations! (sorry for our short delayed feedback!)
>
> We would like to claim that **our method is essentially different from the baselines**. The components you have mentioned ("what kind of success feedback is provided", "high frequency") are the **unique advantages of our framework derived from the LLM-reasoned constraints, rather than some additional "system components"**. Please allow us to explain them in detail:
>
> -  "What kind of success feedback is provided" component
>
> It might be straight-forward that QA stands as a more compact and efficient form of feedback, **but implementing this idea is non-trivial which requires introducing more seamless form of cooperation between LLMs and VLMs, constituting one of the core novelties of our method.**
>
> In our framework, the LLM first reason the specific constraints, and then the VLM is queried with these specific constraints in the form of question and answer (QA) to get more accurate feedback. However, as for Inner Monologue, the agent cannot directly use QA for feedback and only use scene descriptor for feedback because the agent **does not know which part to focus on or which question should be asked** without the LLM guidance and reasoning. In the limitation part (section 5) of the original Inner Monologue paper[1], the author admits their shortage that **they need to manually design questions** to gather specific types of feedback, which are un-scalable and handcrafted, so we labeled it as Inner monologue-Oracle. In contrast, DoReMi can eliminate the need for human-designed questions(feedback types) thanks to the LLM reasoned constraints. **In summary, without the novel mechanism we proposed, it's impossible for Inner Monologue to obtain the same form of feedback automatically.**
>
> - "Frequency of re-planning" component
>
> The form of QA with binary answers compared to scene description is able to enable high-frequency, which is another advantage enabled by LLM-generated constraints. **For the QA format(constraint check), it's straightforward to apply an if-else logic to determine the need for replanning.** In contrast, **scene descriptions require a more time-consuming analysis of semantics and global information to make a replanning decision.** Additionally, **the longer the text output by the VLMs, the more time the processing takes.** In comparison to QA, scene descriptions can take significantly longer – approximately 20 to 50 times more – due to their substantially longer output token lengths.
>
> [1] Huang W, Xia F, Xiao T, et al. Inner monologue: Embodied reasoning through planning with language models[J]. arXiv preprint arXiv:2207.05608, 2022.
>
> We hope the above answer can solve your concern and we are always ready to provide any further explanations! :)

---

> ### Author Response · Authors · 2023-11-23
> **Dear Reviewer nVnU: Welcome to see our new baseline comparison!**
>
> Dear Reviewer nVnU,
>
> In addition to the analysis above, we also conducted new experiments to better verify our unique advantage of precise feedback. We ran a baseline in which Inner Monologue used the same fine-tuned VLM as scene descriptors. As mentioned above, Inner-Monologue can not determine which part of the current scene to focus on or which question to ask. We found that without the help of LLM guidance (in the form of constraints), **Inner-Monologue still performs poorly, even with the fine-tuned VLM.** We initiated these experiments as soon as we received your message. Due to the limited time, we are presenting the results from the most challenging prepare-food environments below.
>
> |                                                           | Inner Monologue | *Inner Monologue-FT* | DoReMi-FT     |
> | --------------------------------------------------------- | --------------- | ------------------ | ------------- |
> | Prepare food with pick failure 0.1 and random drop p=0.0  | 83$\pm$4        | 86$\pm$6           | **96$\pm$3**  |
> | Prepare food with pick failure 0.1 and random drop p=0.02 | 56$\pm5$        | 60$\pm7$           | **93$\pm 5$** |
> | Prepare food with pick failure 0.1 and random drop p=0.04 | 21$\pm7$        | 28$\pm9$           | **91$\pm 6$** |
>
>
> As this discussion period draws to a close, we would like to express our deepest gratitude for your feedback. **We must let you know that we really love the amazing points you raised and thoroughly enjoyed the engaging discussions with you.** We sincerely hope that our responses have satisfactorily addressed your questions. :)
>
> Warm regards,
>
> Authors

---

### Official Review · Reviewer_fXV5 · 2023-10-30

**Soundness:** 3 good
**Presentation:** 3 good
**Contribution:** 3 good
**Rating:** 8
**Confidence:** 3

**Summary:**

The paper introduces DoReMi, a system that uses large language models (LLMs) for high-level planning in robotics and to generate constraints indicating execution misalignments. Vision language models (VLMs) then continuously check for these misalignments. This allows for timely adjustments when there's a deviation from the plan, improving task success rates and completion times in robotic tasks.

**Strengths:**

1. The paper is articulately composed.
2. The figures and the supplementary video enhance comprehension.
3. The central idea is both compelling and clear-cut.
4. Numerous experiments validate the benefits of the proposed approach.

**Weaknesses:**

1. In more intricate environments like "prepare-food," the zero-shot performance of the VLM does seem to underperform, as evidenced by the low success rates in Tables 4-5 of the appendix. There's a concern about the potential for incorrect VLM results to misguide the LLM, leading it to continually replan and squander time. However, as noted in Table 2, DoReMi still surpasses the baselines. Could the authors provide insight into this discrepancy?

2. Section 4.4's theoretical analyses raise some questions for me. Theorem 1 appears to be a direct consequence of the assumed Poisson distribution and doesn't particularly underscore the enhancements brought about by the proposed approach. I wonder if a more constructive approach would be for the authors to hypothesize about the VLM's accuracy, which might allow for an estimation of time savings.

**Questions:**

Please see the content in Weakness.

---

> ### Author Response · Authors · 2023-11-13
> **Response to Reviewer fXV5**
>
> We are genuinely grateful for the time and effort you put into reviewing our paper, and we're especially thankful for your high rating of our manuscript. We trust that the updated PDF and our detailed response will address and resolve any concerns you may have.
>
> ---
>
> > Q1: In more intricate environments like "prepare-food," the zero-shot performance of the VLM does seem to underperform, as evidenced by the low success rates in Tables 4-5 of the appendix. There's a concern about the potential for incorrect VLM results to misguide the LLM, leading it to continually replan and squander time. However, as noted in Table 2, DoReMi still surpasses the baselines. Could the authors provide insight into this discrepancy?
>
> **RESPONSE:** Thank you for your perceptive comment! We looked through the rendered video and found a couple of insights into why DoReMi still surpasses the baseline with zero-shot VLM in prepare-food tasks:
>
>
> - First of all, it's observed that the inaccuracies of the VLM are more likely to cluster around specific trajectories rather than being randomly distributed. In a zero-shot setting, the VLM's accuracy varies: it is less precise with certain objects and scenarios while maintaining accuracy in others.
>
> - Secondly, while zero-shot VLM might make errors in checking specific constraints generated by LLM, the baseline's open-ended scene description could introduce large ambiguity and miss crucial information, potentially leading to even worse performance. This comparison is illustrated in Figure 3 of the manuscript.
>
> We hope these insights can solve your concern.
>
> ---
>
>
> > Q2: Section 4.4's theoretical analyses raise some questions for me. Theorem 1 appears to be a direct consequence of the assumed Poisson distribution and doesn't particularly underscore the enhancements brought about by the proposed approach. I wonder if a more constructive approach would be for the authors to hypothesize about the VLM's accuracy, which might allow for an estimation of time savings.
>
> **RESPONSE**:
> Thank you for your inspirational comments. Given our assumption that misalignments (accidents) occur **randomly and independently throughout the entire execution period**, the Poisson process emerges as the most appropriate model for our scenario. Following your suggestion, we assume the detector has probability $p_d$ to detect the misalignment. With this new assumption, we derive the new results of the potential time savings and success rate improvement.
>
> The updated time-saving $t_s$ and success rate improvements $P_s$ are as follows.
>
> $$\mathbb{E}(t_s)=\frac{p_d \lambda (\mu^2+\sigma^2)}{2}-\lambda p_d\Delta t, \mathbb{E}(P_s)\approx p_d \lambda\mu-\frac{(2p_d-p_d^2)\lambda^2(\mu^2+\sigma^2)}{2}$$
>
> The insight is that as misalignment happens independently, a detector with probability $p_d$ will decrease the misalignment density from $\lambda$ to $(1-p_d)\lambda$ and we can follow the original proof to derive the new equation. The detailed modifications and proof have been added to Section 4.4 and Appendix A.
>
> ---
>
> Thank you again for your time and efforts in reviewing our paper! We hope our response above can solve all your concerns. We show our deepest thanks for your recognition and support of our paper and we are always ready to answer any of your questions!

---

> ### Author Response · Authors · 2023-11-22
> **Thank you for your endorsement of our paper!**
>
> Dear Reviewer fXV5:
>
> We would like to express our profound gratitude for your endorsement of our paper. As we come to the end of the rebuttal phase, we hope we have answered all your questions and shown the improved quality of the paper. Once again, thank you for your time and effort in reviewing our paper!
>
> Best wishes,
>
> Authors

---

> > ### Comment · Reviewer_fXV5 · 2023-11-22
> >
> > Thanks for the response from the authors. Considering the overall quality of the paper, I hold my original score.

---

> > > ### Author Response · Authors · 2023-11-23
> > > **Dear Reviewer fXV5: Thank you for your endorsement!**
> > >
> > > Dear Reviewer fXV5,
> > >
> > > We are delighted to receive your response, and are also glad to learn that our answers have addressed your questions. Have a wonderful Thanksgiving. : )
> > >
> > > Warm regards,
> > >
> > > Authors

---

### Official Review · Reviewer_VZHX · 2023-11-01

**Soundness:** 2 fair
**Presentation:** 2 fair
**Contribution:** 2 fair
**Rating:** 5
**Confidence:** 4

**Summary:**

This paper presents an approach to use LLMs with low-level skills to perform tasks. Specifically, this paper considers cases where the low-level skills fail to perform the sub-task successfully and a separate VLM is used to monitor success and failure of each skill. Once failure is detected (by repeatedly checking at some fixed frequency) the method calls the LLM for replanning from the given state.

**Strengths:**

The main contribution of this paper is to detect failures in skill execution using a VLM and using it to trigger LLM replanning. While previous methods do not trigger failure modes and thus have to wait for low-level skill execution to finish before replanning, this paper adds a VLM based failure detection for replanning.

**Weaknesses:**

I don’t think the main contributions of the paper are very significant. Infact, it is a very simple extension of previous approaches which assume no replanning.   Moreover since VLMs have been used for success/failure detection previously, the use of VLMs to detect skill failures is not really novel and the only component that this paper adds is a very trivial LLM replanning step.

Overall, this paper is basically an implementation of termination conditions (as defined/used in the options literature) with a VLM and a high-level LLM planner.

I also have some other concerns with the paper.

**Using VLM for skill failures:** Why do we have to use a separate VLM (finetuned for skill failures). Since in many cases the skills are being learned why should we not use the skill value function to detect success or failures. This is useful because in the current setting success/failure is completely separate from policy execution. However, in many scenarios when we learn policies we can detect if we are very far from the skill distribution (atleast to some extent).

**Disconnect between constraint generation and skills:** Ideally constraints should depend upon the skills in some way. For instance, as is used in classical symbolic planning literature. However, this approach takes skills as pure black boxes and the constraints are generated simply by the LLM without really considering for the skill. Infact, this implicit shared knowledge between skills and the high-level planner is actually quite important and in this work it is being provided through prompting but that is not really a scalable or general approach.


**Frequency of VLM:** Since the paper uses VLMs for failure detection and proper execution it relies on the inference time of this VLM for fast execution. However, large VLM models (e.g. billions of parameters) will not be fast.

**Assumption of recovery skills:** This work implicitly assumes that all recovery skills are available for the planner. However, this is an unrealistic assumption and simply due to the very toy setups used in this paper.

Overall, replanning is very important. However, this paper takes an overly simplistic approach towards this problem. It doesn’t tackle any of the more challenging and interesting cases and thus the overall contributions are really shallow and fail to provide a significant contribution.

### Update

After rebuttal I have increased the score to 5, I am still on the fence due to the limited technical novelty of the paper. I would be more enthusiastic for the paper, if any of the experiments were carried in more realistic/challenging settings.

**Questions:**

Please see above.

---

> ### Author Response · Authors · 2023-11-13
> **Response to Reviewer VZHX (Part 1/2)**
>
> Thank you for the time and effort you dedicated to reviewing our paper! We would like to provide the following explanations to resolve some misunderstandings. We deeply appreciate that you could reconsider our manuscript with these responses.
>
> ---
>
> **About the contribution and novelty**
>
> > **REVIEWER:** I don’t think the main contributions of the paper are very significant, it's a very simple extension of previous approaches which assume no replanning.
> >
> > Overall, this paper is basically an implementation of termination conditions (as defined/used in the options literature) with a VLM and a high-level LLM planner.
>
> **RESPONSE:** We are afraid misunderstandings exist regarding this paper's contents. We would like to clarify our method is not "a simple extension for termination conditions" or "LLM-replanning". Instead, we propose a novel LLM grounding framework, which utilizes the LLM's reasoning capabilities to generate constraints and employs VLM for constraint detection. This cooperation between LLMs and VLMs in our 'DoReMi' approach results in more accurate and frequent feedback, thus significantly enhancing the re-planning and recovery processes in decision-making tasks. The novelty and advantages of our frameworks are summarized as follows:
>
> * LLM for both plan and constraint generation: **To the best of our knowledge, we are the first to adopt LLM to generate constraints for skills automatically in decision-making tasks**, leveraging LLMs' common-sense reasoning and in-context learning abilities.
>
> * More precise feedback: The VLM can focus on specific constraints proposed by the LLM and monitor the low-level execution, which results in more precise feedback than open-ended scene captioning (visualized in Figure 3).
>
> * More frequency feedback and immediate recovery: The VLM only picks binary answers with short tokens, which leads to frequent feedback.
>
> Detailed descriptions of our framework can be found in sections 4.2 and 4.3. We sincerely wish our claims can solve this misunderstanding and we are always willing to add more clarifications if needed!
>
> ---
>
> > **REVIEWER:**  the paper is a very simple extension of previous approaches which assume no replanning.
>
> **RESPONSE: We are afraid there exist factual errors** in this claim. **In fact, previous works have already considered re-planning and acted as baselines in our paper (Inner monologue[1]).** The advance and effectiveness of our methods lie in how to acquire more precise and frequent feedback, which leads to immediate recovery, as we mentioned above. Experiments in various types of environments and robots further verified the effectiveness of our methods.
>
> [1] Huang W, Xia F, Xiao T, et al. Inner monologue: Embodied reasoning through planning with language models[J]. arXiv preprint arXiv:2207.05608, 2022.
>
> ---
>
> > **REVIEWER:**  Moreover, since VLMs have been used for success/failure detection previously, the use of VLMs to detect skill failures is not really novel and the only component that this paper adds is a very trivial LLM replanning step
>
> **RESPONSE:** We totally agree that VLMs have been used for success/failure detection previously, and we also mentioned them and discussed their shortage in related works of **section 3**. Instead of a success detector, We would like to clarify **we use VLMs in a more general way to check the constraints reasoned by LLMs**, which results in more precise and frequent feedback. The precise feedback further helps LLM to effectively re-plan. Experiments also verified our collaborative approach between LLMs and VLMs is much better than just success detectors.
>
>
> ---
>
> **Why do we use VLM for skill failures but not value functions?**
>
> **RESPONSE:** Thank you for your insightful comment! We would like to provide some reasons why we use VLM instead of value functions.
>
> - First, in many situations, low-level skill failures are rooted in the imperfectness of the skill or, equivalently, the skill value function. In addition, value functions are likely to be inaccurate in out-of-distribution states[1,2], which may fail to detect the accident.
>
> - Second, it may be difficult for low-level skills to consider all types of constraints. For example, locomotion skills mainly focus on the legs of the robots, and it is difficult for the locomotion policy to detect whether the objects in the hands are dropped.
>
> - Third, many skills are driven through model-based control or imitation learning, in which access to value functions might not be available. This circumstance underscores the necessity for an external detector to identify any violations of constraints during the execution process.
>
> [1] Kumar A, Zhou A, Tucker G, et al. Conservative q-learning for offline reinforcement learning[J]. Advances in Neural Information Processing Systems, 2020, 33: 1179-1191.
>
> [2] Yu T, Thomas G, Yu L, et al. Mopo: Model-based offline policy optimization[J]. Advances in Neural Information Processing Systems, 2020, 33: 14129-14142.

---

> ### Author Response · Authors · 2023-11-13
> **Response to Reviewer VZHX (Part 2/2)**
>
> **The connection between constraint generation and skills**
>
> > **REVIEWER:**  The constraints are generated simply by the LLM without really considering the skill
>
> **RESPONSE:**  We would like to clarify that **our constraint is reasoned by LLM which takes into account the current skill and its historical context, as detailed in Section 4.2 of our manuscript.** Subsequent to the LLM's selection of the next skill, we further utilize the LLM to generate specific constraints for the current skill, taking into account the historical steps. This leverages LLM's capabilities in common-sense reasoning and in-context learning. It's worth mentioning that the same skill could yield in distinct constraints when placed in different historical contexts.
>
> > **REVIEWER:**  In fact, this implicit shared knowledge between skills and the high-level planner is actually quite important, and in this work, it is being provided through prompting but that is not really a scalable or general approach.
>
> **RESPONSE:** We respectfully offer an alternative perspective. Prior research has demonstrated the remarkable planning capabilities of LLMs when provided with suitable prompts [1]. In our work, we further expand upon this by leveraging LLM's common-sense reasoning ability to automatically generate constraints for unseen skills. Given LLM's remarkable ability in understanding and reasoning, we think our approach is more scalable and general than previous symbolic methods and rule-based planners that depend on manual design or specific domain knowledge.
>
> [1] Ahn M, Brohan A, Brown N, et al. Do as i can, not as i say: Grounding language in robotic affordances[J]. arXiv preprint arXiv:2204.01691, 2022.
>
> ---
>
> **Frequency of VLM**
>
> **RESPONSE:** We kindly remind that **we analyzed the time cost of VLM detectors (with 5B/13B parameters) in Section 4.3 of our manuscript.** One of the most important factors that may slow down the VLM's inference is the long output token length since tokens are generated auto-regressively. However, in our setting, we alleviate this issue by asking the VLM detector only to provide a binary answer in {"Yes", "No"}, which consists of only one or two tokens, thus leading to a very small query cost (less than 0.1s). This is enough for high-frequency detection with an interval of $\Delta t=0.2s$.
>
> ---
>
> **Assumption of recovery skills**
>
> **RESPONSE:** In our paper, **we provide the most commonly used robot skill sets and consider as many types of disturbances as possible** in various types of environments and robot settings. The robot skill sets contain the most common skills including picking objects, transporting objects, placing objects, and locomotion. The disturbances are also of various types, including failure in picking, random drop during transportation, noise in placement position, unexpected obstacles on the way, etc. And we verified the effectiveness of our framework under various environments, tasks, and robots.
>
> We totally agree that there exist more complex situations that are very hard to recover and more powerful low-level skills are needed. The main contribution of our works lies in proposing a general and scalable LLM-grounding framework which has the potential to work in more complicated scenarios.
>
> ---
>
> Thank you for your time in reviewing our paper again. We deeply appreciate that if you could carefully reconsider our manuscript again to resolve these misunderstandings. We are always ready to answer any of your further questions!

---

> ### Author Response · Authors · 2023-11-19
> **Dear Reviewer VZHX:**
>
> Dear Reviewer VZHX:
>
> We hope that our comprehensive responses have sufficiently clarified the issues you raised.
>
> Considering that our methods have been recognized as compelling [FXV5], innovative [WGrL], and significant [nVnU] by the other three reviewers, we strongly wish you could re-evaluate our manuscript with these perspectives in mind.
>
> Please know that we are fully prepared to respond to any additional queries or concerns you may have.
>
> Sincerely,
> The Authors

---

> > ### Comment · Reviewer_VZHX · 2023-11-19
> > **Thank you for your comments**
> >
> > >> The advance and effectiveness of our methods lie in how to acquire more precise and frequent feedback,
> >
> > I understand that there is more immediate feedback (I know that InnerMonologue replans after the end of skill execution) but that is only a function of checking repeatedly in an almost brute force way and that is the main difference. However, I am not sure if this really consists as some strong technical contribution.
> >
> > >> we use VLMs in a more general way to check the constraints reasoned by LLM.
> >
> > Again I agree with the authors that constraint generation has not been done before. Although, the constraints are just binary detectors with a more fine-grained structure than the final task success. It’s also unclear if generating these constraints requires constraints from all the previous tasks. From the supplementary prompts, it seems that the authors do provide these intermediate values but for more complex tasks where there could be (n choose 2) constraints their approach may scale poorly.
> >
> > I will consider updating the score. Unfortunately, I still don't think there is a lot to learn from this paper. The idea of constraint generation and immediate feedback at every delta step is really more of an application of an LLM to this problem instead of solving some core problem. The experiments also further use very toy scenarios where robust manipulation skills (e.g. pick/place) are artificially made worse by adding noise. While in principal I agree that this is fine for some experiments, there should also then be experiments where these problems realistically occur and the paper provides a solution to. Overall, I still think this an application paper and does not provide any sufficient new insights or interesting results.

---

> ### Author Response · Authors · 2023-11-20
> **Further response and thank you for your time! (Part 1/2)**
>
> We are delighted to receive your response and thank you for taking the time to write to us! To address your points, we would like to further offer the following explanations:
>
>
> ---
>
> > **Comment 1:** I understand that there is more immediate feedback (I know that InnerMonologue replans after the end of skill execution) but that is only a function of checking repeatedly in an almost brute force way and that is the main difference.
>
> **Response:** First, we would like to clarify this kind of real-time detection and feedback is **almost impossible in the previous work of LLM-based planners**. As we mentioned in section 3, Inner monologue can not obtain such high-frequency and highly precise feedback: the success detector only checks the success/failure of the current skill upon finish; human feedback can not be highly frequent; passive scene descriptors tend to be inaccurate. Our novel framework enables seamless cooperation of LLMs and VLMs by generating and detecting constraints, making high-frequency and accurate feedback possible!
>
> Moreover, feedback frequency is only one aspect of improvement. **Another (even more) critical difference is we enable more precise feedback by forcing VLMs to focus on the LLM-generated constraints**. As shown in Figure 3, open-ended scene descriptors of Inner Monologue may be ambiguous. In contrast, LLM can help reason constraints and identify which aspects of observations are critical, thereby improving the precision of feedback. The addition of the "periodic-replan" baseline further underscores the importance of precise feedback. Essentially, without sufficiently accurate feedback, increasing its frequency is futile, as it may lead to the consistent reception of erroneous information and subsequent errors. Thus, both precision and high frequency are crucial for immediate recovery, and our framework has enhanced both aspects.
>
> As for the notion of the "brute force way", we **admit that higher computational is needed to enable higher-frequency feedback. The principle of 'no free lunch' applies here; higher-frequency feedback may necessarily lead to more computational resources**. However, these computational resources are reasonable. As explained in section 4.2, we ask the VLM to choose binary answers with very short tokens for rapid feedback. For our tested 5B and 13B models, this approach allows each query to be processed in less than 0.1 seconds on standard hardware (e.g. a single RTX 3090 card).
>
> We sincerely hope these detailed explanations can solve your concern!
>
> ---
>
> > **Comment 2:** Again I agree with the authors that constraint generation has not been done before. Although, the constraints are just binary detectors with a more fine-grained structure than the final task success. It’s also unclear if generating these constraints requires constraints from all the previous tasks. From the supplementary prompts, it seems that the authors do provide these intermediate values but for more complex tasks where there could be (n choose 2) constraints their approach may scale poorly.
>
> **Response:**  Thank you for recognizing our novelty of constraint generation. As detailed in Sections 4.1 and 4.2 of our manuscript, we effectively harness the **few-shot in-context learning capabilities of LLMs** to generate both the plan and corresponding constraints. Building on prior research findings [1, 2, 3], LLMs can finish challenging planning problems with few-shot examples. In addition to the plans, we further provide some constraints for tasks as examples. This approach **implicitly invokes the reasoning ability of LLM to generate constraints for brand-new tasks** [4]. Our experiments also verified LLM can perform both planning and constraint generation in novel tasks with a diverse range of object types and quantities. With the rapid and ongoing advancements in LLM technology, our framework stands to gain substantially from these developments, enhancing its capability to address increasingly complex tasks with a significantly reduced need for human supervision. Last but not least, it's essential to mention that our approach supports **generating and detecting multiple constraints** for one skill, further maintaining the intersection of constraints, the "Stack blocks in order" task clearly illustrates this aspect.
>
> [1] Ahn M, Brohan A, Brown N, et al. Do as i can, not as i say: Grounding language in robotic affordances[J]. arXiv preprint arXiv:2204.01691, 2022.
>
> [2] Zeng A, Attarian M, Ichter B, et al. Socratic models: Composing zero-shot multimodal reasoning with language[J]. arXiv preprint arXiv:2204.00598, 2022.
>
> [3] Huang W, Abbeel P, Pathak D, et al. Language models as zero-shot planners: Extracting actionable knowledge for embodied agents[C]//International Conference on Machine Learning. PMLR, 2022: 9118-9147.
>
> [4] Brown T, Mann B, Ryder N, et al. Language models are few-shot learners[J]. Advances in neural information processing systems, 2020, 33: 1877-1901.

---

> ### Author Response · Authors · 2023-11-20
> **Further response and thank you for your time! (Part 2/2)**
>
> > **Comment 3:** The idea of constraint generation and immediate feedback at every delta step is really more of an application of an LLM to this problem instead of solving some core problem.
>
> **Response:** We firmly believe that our work addresses the fundamental challenges of recovery in long-horizon decision-making tasks by introducing a mechanism to obtain precise and frequent feedback in a scalable and automatic way. As has been discussed many times before, our work effectively tackles the problem of inaccuracy, impracticality, and low frequency in prior works (e.g., Inner Monologue) by leveraging VLMs to detect binary constraint violation at high frequency. While prior works need to hand-craft success detectors and detection criteria, we automatically generate constraints with LLMs. With these advantages, we achieve high-precision, high-frequency, and automatic constraint violation detection and recovery, which are all critical aspects of recovery in long-horizon decision-making tasks. Consider an extreme scenario where perfect and highly frequent feedback is available; in such a case, it would be significantly easier for robots to replan and recover. We aim to harness LLMs to enhance both the precision and frequency of feedback, thereby contributing to advancements in these core problems.
>
> Moreover, we fully acknowledge that our paper is more application-focused. In line with this, **we have deliberately chosen "applications to robotics, autonomy, planning" as the primary area for our submission, believing that delving into applications within these specific subfields is of considerable importance.** Considering the growing sophistication of LLMs, employing their reasoning abilities to solve complex tasks in these domains seems to be a promising and forward-looking approach. This perspective is supported by numerous papers that have ventured into exploring this direction.
>
>
> ---
>
> > **Comment 4:** The experiments also further use very toy scenarios where robust manipulation skills (e.g. pick/place) are artificially made worse by adding noise. While in principal I agree that this is fine for some experiments, there should also then be experiments where these problems realistically occur and the paper provides a solution to.
>
> **Response:** Firstly, we would like to point out that **random drop, pick failure, place noise, and unexpected obstacles are all common-existing realistic real-world disturbances.** Artificially adding noise to perfect low-level skills are just clever implementations to best present these real-world disturbances in simulators, which **are by no means toy scenarios**. To thoroughly evaluate the performance of our method under all these realistic disturbances and DoReMi's generalization potential, in our experiments, we meticulously selected a wide range of robotic tasks, encompassing locomotion and manipulation. Moreover, our experiment setting is also aligned with the acknowledgment that the low-level controllers we use in the real world are not flawless but always with random noises. The added noise also verifies that DoReMi has the potential to support various low-level controllers.
>
> ---
>
> We sincerely appreciate your reply. If you have any concerns, please feel free to point them out. We are always ready to answer your questions. It's worth pointing out that we would like to express our wish for your support in terms of scoring our work. Thank you for considering our request.

---

> ### Author Response · Authors · 2023-11-22
> **Dear Reviewer VZHX:  Thanks for your valuable time and efforts in reviewing our paper!**
>
> Dear Reviewer VZHX
>
> Thanks for your valuable time and efforts in reviewing our paper! We understand that you may be rather busy during this discussion period. As the rebuttal stage is drawing to a close within a day, we would like to kindly request your feedback on our responses. We are happy to discuss with you in detail if you have additional comments about our paper. :)
>
> Look forward to your reply.
>
> With deepest gratitude,
>
> The Authors

---

### Author Response · Authors · 2023-11-17
**Overall Response**

Dear all reviewers and ACs:

We sincerely appreciate all reviewers’ and AC’s time and efforts in reviewing our paper! We thank you all for the insightful and constructive suggestions, which helped further improve our paper. We have updated our manuscript with editions highlighted in **blue**. Please allow us to summarize our contributions and modifications as follows:

**1. Contributions**

- **Method:** In order to better ground LLM in physical robotic tasks, we propose a novel LLM grounding framework that leverages LLM to reason constraints and VLM to detect these constraints. Through this cooperation between LLMs and VLMs, DoReMi enables more precise and frequent feedback, thus helping re-plan and recover in decision-making tasks. To the best of our knowledge, we are the first to leverage LLM to generate constraints for skills, leveraging LLM's commonsense reasoning ability. The whole framework is compelling, novel, and significant **[Reviewer FXV5, WGrL, and nVnU]**.

- **Experiment:** The benefits of the framework are sufficiently justified with comprehensive quantitative experiments **[Reviewer FXV5 and WGrL]**.

- **Presentation:** The paper is well-written, clear, and articulately composed, and contains many useful diagrams and algorithm blocks that help convey the main messages **[Reviewer FXV5 and nVnU]**.

**2. Modifications:**

- Following Reviewer nVnU's suggestions, we add **a baseline with periodic replan**. The results further verified two essential improvements of DoReMi, including more precise feedback and higher-frequency feedback. Thanks Reviewer nVnU for your insightful comments which further enhance our contribution!

- Following Reviewer nVnU's suggestions, we **count the replanning number and make ablation studies on constraint detect interval**. The results further verified immediate detection leads to a higher success rate and shorter execution time.

- Following Reviewer WGrL's suggestions, we **improved our presentation** by discarding repeated analysis and transferring some explanation from the Appendix to the main paper. We thank Reviewer WGrL's careful feedback and hope these changes can make the paper clearer!

- Following Reviewer fXV5's suggestions, we **consider detector accuracy in our theoretical analysis** and drive the new equations under more practical settings. Thanks Reviewer fXV5's constructive comment which helps improve our theoretical results!

Thank you for your time and effort again! We hope our pointwise responses below could clarify all reviewers' confusion and show the improved quality of our paper. Please let us know if we can provide additional clarifications; we are more than welcome to answer any further questions!

Looking forward to your reply!

Best,

Authors

---

> ### Author Response · Authors · 2023-11-22
> **Dear all reviewers and ACs: At the end of the rebuttal stage**
>
> Dear all reviewers and ACs:
>
> As the end of the rebuttal stage is approaching, we would like to express our sincere appreciation for your invaluable insights and guidance. In the last moments, we are eager to seize every opportunity to address any remaining questions you may have. Please feel free to reach out with any additional inquiries; we are readily available and fully committed to engaging in further discussions and resolving any concerns.
>
> Best regards,
>
> Authors

---

> ### Author Response · Authors · 2023-11-23
> **Welcome any discussion if you want at the last moment!**
>
> Dear all reviewers and ACs:
>
> With less than half an hour remaining in the discussion period, we are still here to have any discussion if you want until the last moment. Your insights and feedback are greatly appreciated, and we are thankful for the effort and time you have dedicated to reviewing our work.
>
> Best regards,
>
> Authors

---

### Meta-Review · Area_Chair_fsji · 2023-12-12

**Metareview:**

The paper proposes a framework (DoReMi) that employs a vision-language model (VLM) to identify instances when the low-level execution of a skill identified by a large language model (LLM)-based planner fails. The VLM operates by continuously checking to see whether LLM-generated constraints are violated. Upon detecting such a failure, the framework halts execution of the current action and queries the LLM to replan from the given state. Evaluations on robot manipulation and humanoid robot domains reveal that the feedback from the VLM improves the success rate and decreases the overall execution time relative to baselines.

The paper was reviewed by four reviewers, each of whom read and responded to the authors' comments, which were extensive and thorough. The paper is quite topical given the significant interest that is being paid to the use of LLMs for robot reasoning and planning. As several reviewers point out, the benefits of being able to identify when the execution of an LLM-suggested skill has failed and immediately replan are clear, a benefit that is supported by the experimental evaluation.

However, as some of the reviewers point out, it is not evident from the paper whether its contributions lie in the proposal of a novel method for identifying and mitigating these failures; in the proposal of a systematic approach that combines existing methods in a novel way or that provides new insights into how such a system may be implemented or how it performs, or in a demonstration of the benefits of preemptive replanning with LLMs. Given that VLMs have previously been used to detect failed (or successful) skill executions, the algorithmic contribution would be the LLM replanning, but as Reviewer VZHX comments, the significance of this contribution is limited. Meanwhile, the fact that IM-Oracle outperforms DoReMi with regards to success rate in several cases raises the question of whether the gains that DoReMi achieves are a result of the ability to prematurely stop low-level skill execution, a capability that would seemingly be easy to add to existing methods as suggested by Reviewer nVnU. In their response, the authors comment that the improvements are a result of being able to provide more precise feedback at higher frequencies, however it is not evident that is achieved via novelties in the framework as opposed to the integration of existing methods (namely VLMs for success/failure detection).

The AC acknowledges the significant effort that the authors put into addressing the questions and concerns that the reviewers initially raised, which involved conducting several new experiments, and their efforts to engage the reviewers in discussion (which the AC also encouraged). Incorporating these results and revisiting the way that the framework is positioned in light of the above comments, will strengthen the paper.

**Justification For Why Not Higher Score:**

While one reviewer recommended Accept, their self-assigned confidence score was lower than that of the other three reviews. In the AC's opinion, this is consistent with their review, which indicated that they are not as familiar with the area as the other reviewers. The AC notes that they are a junior researcher. The AC agrees with the concerns raised by Reviewers VZHX and nVnU regarding the lack of algorithmic novelty and the limited scope of the contributions in light of existing work in this area.

**Justification For Why Not Lower Score:**

N/A

---

### Decision · Program_Chairs · 2024-01-16

Reject